# From Differential Privacy to Bounds on Membership Inference: Less can be More

**Anvith Thudi**                                                                 *anvith.thudi@mail.utoronto.ca*
*University of Toronto and Vector Institute*

**Ilia Shumailov**                                                                 *Ilia.Shumailov@cl.cam.ac.uk*
*University of Toronto and Vector Institute*

**Franziska Boenisch**                                                           *franziska.boenisch@vectorinstitute.ai*
*University of Toronto and Vector Institute*

**Nicolas Papernot**                                                             *nicolas.papernot@utoronto.ca*
*University of Toronto and Vector Institute*

**Reviewed on OpenReview:** *https://openreview.net/forum?id=daXqjb6dVE*

## Abstract

Differential Privacy (DP) is the de facto standard for reasoning about the privacy of a training algorithm. Yet, learning with DP often yields poor performance unless one trains on a large dataset. In this paper, we instead outline how training on *less* data can be beneficial when we are only interested in defending against specific attacks; we take the canonical example of defending against membership inference. To arrive at this result, we first derive (tight) bounds on the success of all membership inference attacks. These bounds do not replace DP, rather they introduce a complementary interpretation of a DP algorithm's ability to defend against membership inference specifically. Because our bound more tightly captures the effect of how training data was selected, we can show that decreasing the sampling rate when constructing the training dataset has a disparate effect on the bound when compared to strengthening the DP guarantee. Thus, when the privacy protection we care about is defending against membership inference, training on less data can yield more advantageous trade-offs between preventing membership inference and utility than strengthening the DP guarantee. We empirically illustrate this on MNIST, CIFAR10 and SVHN-extended.

## 1 Introduction

Differential Privacy (DP) is employed extensively to reason about privacy guarantees (Dwork et al., 2006; Dwork, 2008). Recently, DP started being used to give privacy guarantees for the training data of deep neural networks (DNNs) learned through a noisy stochastic gradient descent (SGD) (Abadi et al., 2016). However training DNNs with DP has come at a cost to performance. The emerging paradigm to resolve this tension between performance and privacy is to get access to more private data and/or do pretraining on public data, i.e., data whose privacy we need not protect (Tramèr & Boneh, 2020; De et al., 2022; Mehta et al., 2022). Here, we take a different direction. **Perhaps surprisingly, we show that training on *less* data can be beneficial to the privacy vs. performance trade-off when one cares about defending against a specific class of privacy attacks.** In particular, constrained to a fixed privacy guarantee we obtain models that become more accurate by training on a smaller subset of the available training data.

This is because differential privacy provides guarantees that hold for many attacks, yet our primary concern may only be preventing a subset of those attacks. Put simply, DP often leads to a characterization of

privacy that is too conservative. We show that a defender interested in preventing only certain forms of privacy leakage, i.e., certain attacks, is well served by deriving bounds tailored to these specific attacks. Such bounds can surface new variables undermining the success of those specific privacy attacks, and hence new rigorous defense mechanisms for those attacks.

We take the canonical example of a membership inference (MI) attack. This is the most practical attack devised against the privacy of training data to this date, as evidenced by the line of work initiated by Shokri et al. (2017). The goal of the MI adversary is to predict whether or not a model used a particular datapoint for training. Defending against membership inference is a common goal in prior work. Approaches either fall into the category of obtaining differential privacy (in a way that is agnostic to the threat of membership inference) or deploying heuristics that forgo differential privacy to mitigate known membership inference attacks (Jia et al., 2019; Nasr et al., 2018). The former introduces a large cost in peformance because the guarantee is too conservative as outlined above. The latter has typically better performance but is not rigorous and thus introduces an arms race between attackers and defenders.

**We propose a change in paradigm: we derive bounds on the success of *any* membership inference adversary that more *tightly* capture non-DP factors.** That is, unlike past bounds that exclusively study the connection between DP and MI, we ask how other factors underlie the success of attacks when one has trained with DP. However, unlike heuristic defenses to membership inference, we provide *guarantees* that bound the success of *any* membership inference adversary; our bound does *not* introduce an arms race between attackers and defenders. In other words, we obtain the best of both classes of approaches considered thus far.

Our **first contribution** is to derive a bound on the success of any membership inference adversary when the model was obtained with a differentially private training algorithm. Specifically, we bound the positive accuracy (*i.e.* precision) of any membership inference: when the adversary predicts a point is a member of the dataset, how likely is it that they are correct? The key novel insight for our bound is that a membership inference adversary only ever has a finite set of points they suspect were used during training. This allows us to apply a counting lemma. We show that our bound is tight up to the looseness of the DP guarantee provided by the training algorithm. This means our bound cannot be improved further without improving the DP guarantee itself or replacing it by another guarantee.

Our **second contribution** is to observe that our bound surfaces a dominant factor in the success of membership inference attacks: how the training set was sampled. This factor has an even larger effect on the bound than the strength of the DP guarantee the model was obtained with. It suggests a new paradigm to alleviate the previously observed tension between defending against membership inference (MI) by training with DP and training performative models. Concretely, as our bound better describes the effect of different regimes of sampling training data, we are able to demonstrate the effectiveness of subsampling training data as a form of privacy amplification specific to defending against MI. By subsampling the training data, we are able to obtain increased privacy in the form of increased robustness to MI that outweighs the degradation in performance that results from training on a smaller dataset. **As a result, we surprisingly find that training on less data can yield more advantageous trade-offs between privacy and performance —when one is primarily concerned about limiting the success of membership inference.**

We illustrate this phenomenon on MNIST, CIFAR10 and SVHN-extended. Notably, we confirm that this theoretical outweighed benefit of subsampling empirically translates to increased robustness to state-of-the-art membership inference attacks (Carlini et al., 2022a). As predicted by inspecting our bound, we also confirm experimentally that training with less data can result in a more accurate model when one is given a fixed target bound (i.e., a maximal MI precision they can tolerate). We attach our code in the supplementary material and will open-source it upon publication.

## 2 Background

### 2.1 Differential Privacy

Differential privacy (DP) (Dwork et al., 2006) bounds how different the outputs of a function on adjacent inputs can be in order to provide privacy guarantees for the inputs. More formally, a function $F$ is $\varepsilon$-DP if for all adjacent inputs $x$ and $x'$ (*i.e.* inputs with Hamming distance of 1) we have for all sets $S$ in the output space:

$$\mathbb{P}(F(x) \in S) \leq e^{\varepsilon}\mathbb{P}(F(x') \in S). \tag{1}$$

There also is the more relaxed notion of $(\varepsilon, \delta)$-DP which is defined for the same setup as above, but introduces a parameter $\delta \in (0, 1]$, such that $\mathbb{P}(F(x) \in S) \leq e^{\varepsilon}\mathbb{P}(F(x') \in S) + \delta$. Notably, $(\varepsilon, \delta)$-DP is used for functions where it is more natural to work with $\ell_2$ metrics on the input space, having to do with how DP guarantees are obtained.

To achieve DP guarantees, one usually introduces noise to the output of the function $F$. The amount of noise is calibrated to the maximal $\ell_2$ or $\ell_1$ difference between all possible outputs of the function on adjacent datasets (also called *sensitivity*). Significant progress was achieved on minimizing the amount of noise needed for a given sensitivity (Balle & Wang, 2018; Dwork et al., 2010; Kairouz et al., 2015).

Song et al. (2013); Bassily et al. (2014); Abadi et al. (2016) demonstrated a method to make the final model returned by mini-batch SGD $(\varepsilon, \delta)$-DP with respect to its training dataset. This is done by bounding the sensitivity of gradient updates during mini-batch SGD and introducing Gaussian noise to each update. Despite being the de facto approach to learning with DP, the adoption of DP-SGD is still greatly limited because of an observed trade-off between privacy guarantees and model performance . As discussed in the introduction, a common approach to address this trade-off is to seek access to additional data that is public (Tramèr & Boneh, 2020; De et al., 2022). We remark now that there are other DP notions used to analyze DP-SGD that provide tighter analysis, such as Rényi DP Mironov (2017) and f-DP Dong et al. (2019), and leave it to future work to study optimal membership inference bounds under those definitions.

### 2.2 Membership Inference

Membership inference (MI) asks, given a model and a datapoint, can the adversary infer if the given datapoint was used in the training set? Shokri et al. (2017) introduced an MI attack against deep neural networks (DNNs), which leveraged shadow models (models with the same architecture as the target model) trained on similar data. Given these shadow models, they then trained a classifier which, given the outputs of a model on a data point, predicts if the model was trained on that point or not. Since the introduction of this initial attack, the community has proposed several improved variations of the original MI attack (Yeom et al., 2018; Salem et al., 2018; Sablayrolles et al., 2019; Truex et al., 2019; Jayaraman et al., 2020; Maini et al., 2021; Choquette-Choo et al., 2021). These privacy attacks are currently the most practical attacks an adversary can implement against a machine learning model.

### 2.3 Previous Bounds

We are interested in providing bounds on the success of MI attacks against algorithms that provide DP. Initially, prior efforts (Yeom et al., 2018; Erlingsson et al., 2019; Humphries et al., 2020; Mahloujifar et al., 2022) only considered the restricted case where a candidate for MI (i.e., a point that may or may not have been used to train the model) always has a prior 50% probability of being a training datapoint. Sablayrolles et al. (2019) were the first to introduce the prior probability $\mathbb{P}_{\mathbf{x}^*}(1)$ of the datapoint being sampled into the training dataset as a factor of their analysis. They bounded the probability a datapoint $\mathbf{x}^*$ was in fact used to train the model (*i.e.* $\mathbb{P}(\mathbf{x}^* \in D_{train}|S)$ which we will refer to as "positive accuracy" in Section 3) by $\mathbb{P}_{\mathbf{x}^*}(1) + \frac{\varepsilon}{4}$. However this bound (a) assumes that the algorithm is $\varepsilon$-DP, *i.e.* , requires $\delta = 0$, and (b) only loosely describes the impact of $\mathbb{P}_{\mathbf{x}^*}(1)$. We address both of these limitations in our paper.

Indeed, our first step will be to give a bound that more tightly captures the effect of $\mathbb{P}_{\mathbf{x}^*}(1)$: a non-DP factor affecting the bound. We point this out as a key observation of our work will be that varying this sampling

probability has a disparate effect on MI (compared to strengthening the DP guarantee). Furthermore, changing the sampling rate is in fact in the model trainer's control, and so this will also allow us to propose practical mechanisms a defender can use to boost their ability to defeat MI.

We lastly point out that concurrent work by Mahloujifar et al. (2022) improves the bound by Sablayrolles et al. (2019) to $\frac{1}{1+e^{-\varepsilon}}$. This however comes at the expense of reintroducing the assumption that $\mathbb{P}_{\mathbf{x}^*}(1) = 0.5$ (*i.e.* the prior probability of the datapoint being sampled into the training dataset is 50%) and an additional assumption that all other data is fixed. Our bound will match theirs at this sampling rate $\mathbb{P}_{\mathbf{x}^*}(1) = 0.5$, but notably generalizes to having multiple variable datapoints all with different sampling rates. This generalization is crucial for making subsampling a mechanism a defender can use[1]. A more detailed discussion comparing the different aforementioned bounds and their proof techniques in light of the advances made in our paper is deferred to Appendix B.

## 3 Bounding Membership Inference

As highlighted in Section 2.3, past bounds are largely agnostic to, or do not tightly capture, factors influencing membership inference beyond the DP guarantee of the algorithm. We proceed to derive a more general membership inference bound that better incorporates the prior likelihood (before observing the final model) of whether a datapoint was sampled into the training set. The bound we obtain constitutes this paper's first principal result. It is from this improved bound that we later describe how controlling the sampling rate of data into the training set can constitute an effective and desirable defense against membership inference.

### 3.1 Definition of Membership Inference and Notation

We first introduce some notation. In the rest of the paper we assume the training dataset $D$ is a random variable over subsets of a set $\{\mathbf{x}_1, \cdots, \mathbf{x}_N\}$, and use $\mathbf{x}^* \in D$ to denote the event that $D$ contains a particular datapoint $x^*$, formally $D \in \mathfrak{D}_{\mathbf{x}^*}$ where $\mathfrak{D}_{\mathbf{x}^*}$ denotes subsets containing $\mathbf{x}^*$. Later in Section 3.2 we specify the distribution of $D$: a product distribution of i.i.d Bernoulli random variables associated to each datapoint $\mathbf{x} \in \{\mathbf{x}_1, \cdots, \mathbf{x}_N\}$ for some $N$, where we let $\mathbb{P}_{\mathbf{x}}(1)$ denote the probability the bernoulli random variable associated to $\mathbf{x}$ is 1 (representing $\mathbf{x}$ being sampled into $D$). We also use $H$ to denote the training function, which takes as input the training dataset $D$.

The objective of an MI adversary is to predict if the event $\mathbf{x}^* \in D$ occured given only the ouputs of training on $D$, i.e., a function $MI(\mathbf{x}^*, H(D)) \in \{0, 1\}$. More specifically, to every MI adversary, we can associate a set of models (or generally, algorithm outputs) for which it predicts the datapoint $x^*$ was in the training dataset $D$ (i.e., is a *member*). Let us call this set $S$. In fact, the set $S$ completely characterizes the adversary; on $S$ the adversary predicts $\mathbf{x}^*$ was a member, and on $S^c$ the adversary predicts not a member. Hence, now just working with the set $S$, we can abstract what an MI adversary is and state an equivalent definition of membership inference in terms of conditionals. Specifically, we have the following definition of the adversary's "positive accuracy", *i.e.* accuracy when predicting $\mathbf{x}^*$ is in the training set (also called precision or positive predictive value in the literature[2]). For brevity we use $\mathbb{P}(\mathbf{x}^* \in D|S)$ to denote $\mathbb{P}(\mathbf{x}^* \in D|MI(\mathbf{x}^*, M(D)) = 1) = \mathbb{P}(\mathbf{x}^* \in D|H(D) \in S)$, i.e., the conditioning on $S$ denotes conditioning on the MI attack predicting $\mathbf{x}^*$ is a member given $H(D)$.

**Definition 3.1** (MI positive accuracy). The positive accuracy of a MI adversary, associated to the set $S$, is $\mathbb{P}(\mathbf{x}^* \in D|S)$. Similarly, the negative accuracy is $\mathbb{P}(\mathbf{x}^* \notin D|S) = 1 - \mathbb{P}(\mathbf{x}^* \in D|S)$,

For notational simplicity, we will often use $\mathbb{P}(\mathbf{x}^*|S)$ to denote $\mathbb{P}(\mathbf{x}^* \in D|S)$. We explain more about where the randomness is introduced (in particular the probability involved in obtaining a training dataset which the hypothesis test for membership inference acts on) in Section 3.2.

---

[1]A defender would most likely want to preserve the privacy of multiple datapoints simultaneously.

[2]We stick away from using the terms precision and positive predictive value as in the literature they are specific to the "adversary" interpretation of the definition, while it seems to us this abstraction in terms of $S$ may be of more general interest.

### 3.2 The Setting

We now proceed to formalize how an entity samples data into the training dataset, i.e., our prior on whether a datapoint $\mathbf{x}^*$ was used to train a model before observing a specific set of models. The intuition for our formalization is the existence of some finite data superset containing all the data points that an entity could have in their training dataset. Yet, any one of these datapoints only has some probability of being sampled into the training dataset. For example, this larger dataset could consist of all the users that gave an entity access to their data, and the probability comes from the entity randomly sampling the data to use in their training dataset. This randomness can be a black-box such that not even the entity knows what data was used to train. We can then imagine that the adversary knows the larger dataset and tries to infer whether a particular point was used in the training dataset. This setting defines the prior we will assume for the MI attacks we bound, and we now explain the terminology and facts regarding this setup that we will use.

Specifically, let the individual training datasets $D$ be constructed by sampling from a finite countable set where all datapoints are unique and sampled independently, *i.e.* from some larger set $\{\mathbf{x}_1, \cdots, \mathbf{x}_N\}$. That is if $D = \{\mathbf{x}_1, \mathbf{x}_2, \cdots, \mathbf{x}_n\}$ then the probability of sampling $D$ is $\mathbb{P}(D) = \mathbb{P}_{\mathbf{x}_1}(1)\mathbb{P}_{\mathbf{x}_2}(1)\cdots\mathbb{P}_{\mathbf{x}_n}(1)\mathbb{P}_{\mathbf{x}_{n+1}}(0)\cdots\mathbb{P}_{\mathbf{x}_N}(0)$, where $\mathbb{P}_{\mathbf{x}_i}(1)$ is probability of drawing $\mathbf{x}_i$ into the dataset and $\mathbb{P}_{\mathbf{x}_i}(0)$ is the probability of not.

We define $\mathfrak{D}$ as the set of all datasets. Let now $\mathfrak{D}_{\mathbf{x}^*}$ be the set of all datasets that contain a particular point $\mathbf{x}^* \in \{\mathbf{x}_1, \cdots, \mathbf{x}_N\}$, that is $\mathfrak{D}_{\mathbf{x}^*} = \{D \; : \; \mathbf{x}^* \in D\}$. Similarly let $\mathfrak{D}_{\mathbf{x}^*}'$ be the set of all datasets that do not contain $\mathbf{x}^*$, *i.e.* $\mathfrak{D}_{\mathbf{x}^*}' = \{D' \; : \; \mathbf{x}^* \notin D'\}$. Note $\mathfrak{D} = \mathfrak{D}_{\mathbf{x}^*} \cup \mathfrak{D}_{\mathbf{x}^*}'$ by the simple logic that any dataset has or does not have $\mathbf{x}^*$ in it. We then have the following lemma.

**Lemma 3.2.** $\mathfrak{D}_{\mathbf{x}^*}$ *and* $\mathfrak{D}_{\mathbf{x}^*}'$ *are in bijective correspondence with* $\mathbb{P}(D)\frac{\mathbb{P}_{\mathbf{x}^*}(0)}{\mathbb{P}_{\mathbf{x}^*}(1)} = \mathbb{P}(D')$ *for* $D \in \mathfrak{D}_{\mathbf{x}^*}$ *and* $D' \in \mathfrak{D}_{\mathbf{x}^*}'$ *that map to each other under the bijective correspondence.*

*Proof.* Note that for a given $D = \{\mathbf{x}_1, \cdots, \mathbf{x}_n\} \in \mathfrak{D}_{\mathbf{x}^*}$, $D' = D/\mathbf{x}^* \in \mathfrak{D}_{\mathbf{x}^*}'$ is unique (*i.e.* the map by removing $\mathbf{x}^*$ is injective) and similarly for a given $D' \in \mathfrak{D}_{\mathbf{x}^*}'$ $D = D' \cup \mathbf{x}^*$ is unique (*i.e.* the map by adding $\mathbf{x}^*$ is injective). Thus, we have injective maps running both ways which are the inverses of each other. As a consequence, we have $\mathfrak{D}_{\mathbf{x}^*}$ and $\mathfrak{D}_{\mathbf{x}^*}'$ are in bijective correspondence.

Now if the larger set of datapoints is $\{\mathbf{x}_1 \cdots \mathbf{x}_{n-1}, \mathbf{x}^*, \mathbf{x}_n \cdots \mathbf{x}_N\}$ letting $D = \{\mathbf{x}_1 \cdots \mathbf{x}_{n-1}\} \cup \mathbf{x}^*$ and $D' = \{\mathbf{x}_1 \cdots \mathbf{x}_{n-1}\}$ be any pair of datasets that map to each other by the above bijective map, then note $\mathbb{P}(D) = \mathbb{P}_{\mathbf{x}_1}(1)\mathbb{P}_{\mathbf{x}_2}(1)\cdots\mathbb{P}_{\mathbf{x}_{n-1}}(1)\mathbb{P}_{\mathbf{x}^*}(1)\cdots\mathbb{P}_{\mathbf{x}_{n+1}}(0)\cdots\mathbb{P}_{\mathbf{x}_N}(0)$ and $\mathbb{P}(D') = \mathbb{P}_{\mathbf{x}_1}(1)\mathbb{P}_{\mathbf{x}_2}(1)\cdots\mathbb{P}_{\mathbf{x}_{n-1}}(1)\mathbb{P}_{\mathbf{x}^*}(0)\cdots\mathbb{P}_{\mathbf{x}_{n+1}}(0)\cdots\mathbb{P}_{\mathbf{x}_N}(0)$. In particular we have $\mathbb{P}(D)\frac{\mathbb{P}_{\mathbf{x}^*}(0)}{\mathbb{P}_{\mathbf{x}^*}(1)} = \mathbb{P}(D')$. $\square$

Once some dataset $D$ is obtained, we call $H$ the training function which takes in $D$ and outputs a model $M$ represented by a real vector. Recall that $H$ is $(\varepsilon, \delta)$-DP if for all adjacent datasets $D$ and $D'$ and any set of models $S$ in the output space of $H$ (*i.e.* some weights) we have: $\mathbb{P}(H(D) \in S) \leq e^\varepsilon \mathbb{P}(H(D') \in S) + \delta$.

### 3.3 The Bound

We now proceed to use Lemma 3.2 to bound the positive and negative accuracy of MI, as stated in Definition 3.1. This will be done for a training function $H$ that is $(\varepsilon, \delta)$-DP, under the data-sampling setting defined earlier.

From now on we assume that the set $S$ has a non-zero probability to be produced by $H$ given $\mathbf{x}^*$ is in the training set (or not in the training set in the case of negative accuracy bounds). This is sensible as we are not interested in MI positive or negative accuracy for adversaries that never predict positively or negatively respectively.

**Theorem 3.3.** *For any MI attack on datapoint* $\mathbf{x}^*$ *(i.e. with any associated set $S$), given the training process $H$ is $(\varepsilon, \delta)$-DP, we have its positive accuracy is upper-bounded by* $\mathbb{P}(\mathbf{x}^*|S) \leq (1 + \frac{e^{-\varepsilon}\mathbb{P}_{\mathbf{x}^*}(0)}{\mathbb{P}_{\mathbf{x}^*}(1)} - \frac{\delta e^{-\varepsilon}\mathbb{P}_{\mathbf{x}^*}(0)}{\mathbb{P}(S|\mathbf{x}^*)})^{-1}$ *so*

*long as the right-hand side is positive. Similarly we have the negative accuracy is bounded by* $\mathbb{P}(\mathbf{x}^* \notin D|S) \leq$ $(1 + \frac{e^{-\varepsilon}\mathbb{P}_{\mathbf{x}^*}(1)}{\mathbb{P}_{\mathbf{x}^*}(0)} - \frac{\delta e^{-\varepsilon}\mathbb{P}_{\mathbf{x}^*}(1)}{\mathbb{P}(S|\mathbf{x}^* \notin D)})^{-1}$.

*Proof.* The positive accuracy of an adversary $f$ associated to any set $S$ is: $\mathbb{P}(\mathbf{x}^*|S) = \frac{\sum_{D \in \mathfrak{D}_{\mathbf{x}^*}} \mathbb{P}(H(D) \in S)\mathbb{P}(D)}{\sum_{D \in \mathfrak{D}_{\mathbf{x}^*}} \mathbb{P}(H(D) \in S)\mathbb{P}(D) + \sum_{D' \in \mathfrak{D}'_{\mathbf{x}^*}} \mathbb{P}(H(D') \in S)\mathbb{P}(D')}$.

Now using that $\mathbb{P}(H(D') \in S) \geq e^{-\varepsilon}\mathbb{P}(H(D) \in S) - e^{-\varepsilon}\delta$ and $\mathbb{P}(D') = \mathbb{P}(D)\frac{\mathbb{P}_{\mathbf{x}^*}(0)}{\mathbb{P}_{\mathbf{x}^*}(1)}$ we have the denominator is $\geq \sum_{D \in \mathfrak{D}_{\mathbf{x}^*}}(1 + \frac{e^{-\varepsilon}\mathbb{P}_{\mathbf{x}^*}(0)}{\mathbb{P}_{\mathbf{x}^*}(1)})\mathbb{P}(H(D) \in S)\mathbb{P}(D) - \delta e^{-\varepsilon}\sum_{D' \in \mathfrak{D}'_{\mathbf{x}^*}} \mathbb{P}(D')$.

Note that $\delta e^{-\varepsilon}\sum_{D' \in \mathfrak{D}'_{\mathbf{x}^*}} \mathbb{P}(D') = \delta e^{-\varepsilon}\sum_{D' \in \mathfrak{D}'_{\mathbf{x}^*}} \mathbb{P}(D')\frac{\sum_{D \in \mathfrak{D}_{\mathbf{x}^*}} \mathbb{P}(H(D) \in S)\mathbb{P}(D)}{\sum_{D \in \mathfrak{D}_{\mathbf{x}^*}} \mathbb{P}(H(D) \in S)\mathbb{P}(D)}$. So now cancelling the $\sum_{D \in \mathfrak{D}_{\mathbf{x}^*}} \mathbb{P}(H(D) \in S)\mathbb{P}(D)$ in the numerator and denominator, and noting $\sum_{D \in \mathfrak{D}_{\mathbf{x}^*}} \mathbb{P}(H(D) \in S)\mathbb{P}(D) = \mathbb{P}(S|\mathbf{x}^*)$ and $\sum_{D' \in \mathfrak{D}'_{\mathbf{x}^*}} \mathbb{P}(D') = \mathbb{P}_{\mathbf{x}^*}(0)$, we have $\mathbb{P}(\mathbf{x}^*|S) \leq (1 + \frac{e^{-\varepsilon}\mathbb{P}_{\mathbf{x}^*}(0)}{\mathbb{P}_{\mathbf{x}^*}(1)} - \frac{\delta e^{-\varepsilon}\mathbb{P}_{\mathbf{x}^*}(0)}{\mathbb{P}(S|\mathbf{x}^*)})^{-1}$. Note this is only true if the right-hand side of the bound is positive, as the fact $a/b \leq a/c$ if $c \leq b$ (which we used) is only true if $c$ is positive when $a$ and $b$ is positive.

Note that a completely analogous proof proves the negative accuracy bound.

$\square$

**Remark on $\mathbb{P}(S|\mathbf{x}^*)$.** Implicit in Theorem 3.3 is the fact we must accept a probability of failing to meaningfully bound positive accuracy when $\delta \neq 0$, i.e., we need to assume a lower-bound on $\mathbb{P}(S|\mathbf{x}^*)$ when $\delta \neq 0$. Otherwise the expression in Theorem 3.3 can approach $\infty$ as $\mathbb{P}(S|\mathbf{x}^*) \to 0$. Our failure to bound positive accuracy as $\mathbb{P}(S|\mathbf{x}^*) \to 0$ is in fact "necessary" for any bound on positive accuracy, as without a lower bound on $\mathbb{P}(S|\mathbf{x}^*)$ we allow adversaries whose positive accuracy can be arbitrarily close to 1 even when training with $(\epsilon, \delta) - DP$; an example of a class of adversaries whose positive accuracy approaches 1 as we allow $\mathbb{P}(S|\mathbf{x}^*) \to 0$ is given in Appendix A.1. One can interpret this mode of failure as accepting "we may fail for adversaries that would predict positively infrequently if we did train with $\mathbf{x}^*$".

**Remark on tightness of the bounds.** The bounds in Theorem 3.3 are tight if all the differential privacy inequalities we use (specifically $\mathbb{P}(H(D') \in S) \geq e^{-\varepsilon}\mathbb{P}(H(D) \in S) - \delta \; \forall D \in \mathfrak{D}_{\mathbf{x}^*}$) are tight. This is due to those differential privacy inequalities being the only inequalities appearing in the proof. Hence, any future improvement would have to go beyond the DP condition and incorporate the looseness of the DP guarantee on different datasets. This means that our bound is optimal when no such knowledge is provided.

### 3.4 Comparing $\mathbb{P}_{\mathbf{x}^*}(1)$ to the privacy budget $\epsilon$

The bound given by Theorem 3.3 can be reduced by decreasing $\mathbb{P}_{\mathbf{x}^*}(1)$ or the privacy budget $\varepsilon$. Reducing $\mathbb{P}_{\mathbf{x}^*}(1)$ corresponds to "dataset subsampling", *i.e.* downsampling the original training set to a new smaller training set. However, the privacy budget $\varepsilon$ can also be reduced by a sampling strategy: reducing the minibatch sampling rate during DP-SGD is known to reduce the overall $\varepsilon$ by "batch sampling" amplification. So we have that both dataset subsampling and batch sampling provide privacy amplification (for MI) as they decrease our bound. However, is the effect of these two sampling strategies on our bound quantitatively different? As batch sampling acts on our bound by decreasing $\varepsilon$, we will compare the impact on our bound as we vary $\varepsilon$ or $\mathbb{P}_{\mathbf{x}^*}(1)$. In the following, we take $\delta = 0$ for simplicity.

To first get a sense of the impact $\mathbb{P}_{\mathbf{x}^*}(1)$ has on our bound, we plot $\mathbb{P}_{\mathbf{x}^*}(1)$ against the positive MI accuracy bound from Theorem 3.3 in Figure 1 for different $\varepsilon$. The trend we see is that the bound decreases to 0 as $\mathbb{P}_{\mathbf{x}^*}(1)$ goes to 0, and for large $\varepsilon$ this drop happens sharply. However, by decreasing $\varepsilon$ we have the bound converges to $\mathbb{P}_{\mathbf{x}^*}(1)$ which need not be 0.

We now quantitatively compare the effect of $\varepsilon$ to the effect of $\mathbb{P}_{\mathbf{x}^*}(1)$, where the latter appears in our bound as $\mathbb{P}_{\mathbf{x}^*}(0)/\mathbb{P}_{\mathbf{x}^*}(1) = (1 - \mathbb{P}_{\mathbf{x}^*}(1))/\mathbb{P}_{\mathbf{x}^*}(1)$. We can compare the impact of $\varepsilon$ to the one of $\mathbb{P}_{\mathbf{x}^*}(1)$ by looking

at the term $e^{-\varepsilon}\mathbb{P}_{\mathbf{x}}(0)/\mathbb{P}_{\mathbf{x}}(1)$ in the bound given by Theorem 3.3; the goal is to maximize this to make the upper bound as small as possible. In particular, we see that decreasing $\varepsilon$ increases this term by $O(e^{-t})$ whereas decreasing $\mathbb{P}_{\mathbf{x}^*}(1)$ increases this term by $O((1-t)/t)$, which is weaker than $O(e^{-t})$ up to a point, then stronger: we are looking at the order as the variable $t$ decreases. Figure 2 plots this relation, however, the specific values are subject to change with differing constants. Nevertheless what does not change with the constants are the asymptotic behaviours, and in particular we see $\lim_{t\to 0} O((1-t)/t) = \infty$ where as $\lim_{t\to 0} O(e^{-t}) = $ constant.

Thus, we can conclude from our bound that the effects of data sampling and $\varepsilon$ (which encompasses batch sampling) are different. In other words, data sampling presents a non-DP method to reduce MI positive accuracy.

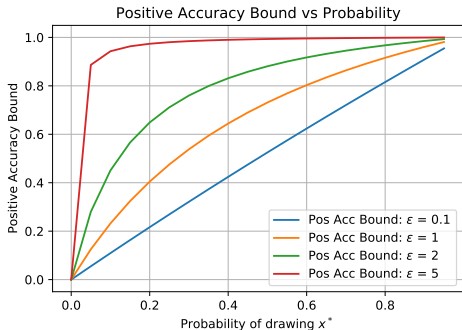

Figure 1: Upper bound on MI positive accuracy from Theorem 3.3, setting $\delta = 0$, as a function of $\mathbb{P}_{\mathbf{x}^*}(1)$

## 4  The MI Sampling Mechanism

We now focus on what will be the key benefit of the bound we derived in Theorem 3.3, the effect of $\mathbb{P}_{\mathbf{x}^*}(1)$ on the success of MI attacks. As we just observed in Section 3.4, the $\mathbb{P}_{\mathbf{x}^*}(1)$ dependence of our bound dominates its $\varepsilon$ dependence. This means that decreasing $\mathbb{P}_{\mathbf{x}^*}(1)$ will impact the bound more than strengthening the differential privacy guarantee (i.e., lowering $\varepsilon$).

We now describe in Section 4.1 how this gives a new mechanism a defender can use to prevent membership inference, which we call the "MI sampling mechanism". Following this, we show in Section 4.2 one can empirically observe the disparate effect of this mechanism in the current state-of-the-art membership inference attack (Carlini et al., 2022a).

This will have explained the usefulness of subsampling the training set for entities that can afford using small sampling rates $\mathbb{P}_{\mathbf{x}^*}(1)$, e.g., have more data than they can train on. However in Section 4.3 we empirically show a favorable trade-off between low sampling rates and performance on MNIST, CIFAR10 and SVHN-extended. That is, we will show training on less data can be beneficial for the privacy vs. performance trade-off on benchmark datasets so long as the entity training the model is interested in only defending against MI attacks.

### 4.1  Usefulness for a Defender

We now explain one course of action a defender can take in light of the privacy amplification for MI given by $\mathbb{P}_{\mathbf{x}^*}(1)$. Note that an upper bound on $\mathbb{P}_{\mathbf{x}^*}(1)$ translates to an upper bound on the relation found in Theorem 3.3. This is as the bound is increasing with $\mathbb{P}_{\mathbf{x}^*}(1)$. Hence one can, in practice, focus on giving smaller upper-bounds on $\mathbb{P}_{\mathbf{x}^*}(1)$ to decrease MI positive accuracy.

A possible approach to this is as follows: assume a user (the defender) is given some sample $D \subset \{\mathbf{x}_1, \cdots, \mathbf{x}_N\}$ drawn with some unknown distribution. In particular, the user does not know the individual probabilities for the points being in $D$ and whether these points were drawn independently. However, assume that the user obtains from $D$ the training dataset $D_{train}$ by sampling any point from $D$ independently with

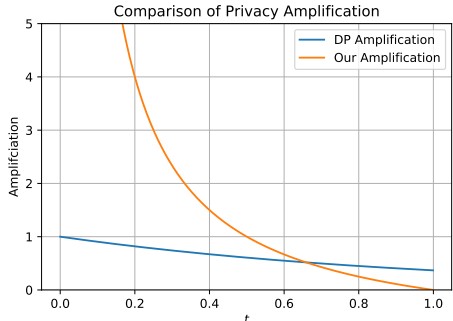

Figure 2: Comparing the DP amplification observed by decreasing batch probability (given by $O(e^{-t})$) to the amplification we observe from decreasing $\mathbb{P}_{\mathbf{x}}(1)$ (given by $O((1-t)/t)$) in Theorem 3.3 setting $\delta = 0$

probability $T$. Our bound then bounds the conditional probability $\mathbb{P}(\mathbf{x}^*|S, D)$ with $\mathbb{P}_{\mathbf{x}^*}(1) = T$. Note that $\mathbb{P}(\mathbf{x}^*|S) \leq \mathbb{P}(\mathbf{x}^*|S, D)$ as $\mathbf{x}^* \in D$, so our upper-bound also bounds $\mathbb{P}(\mathbf{x}^*|S)$. That is, one can use our bound to give upper-bounds for points sampled from unknown (not necessarily i.i.d) distributions if independent sampling (*i.e.* as needed for Lemma 1) is applied *after* drawing from this unknown distribution.

To give a real-world example of this procedure, assume $D$ is obtained by scraping Wikipedia articles, following some protocol that may be deterministic. In this case, without exact knowledge of the sampling probabilities of points, and a guarantee they were independent, we cannot use our bound. However, given $D$, now consider sampling from it specific sentences (i.e., inputs) independently, where each sentence has probability $T$ of being included into the new dataset $D_{train}$. We then train our model using DP with $D_{train}$. We then can apply Theorem 3.3 to bound the positive accuracy any adversary (that makes deterministic decisions given the models returned during training) can have by setting $\mathbb{P}_{x^*}(1) = T$ in the Theorem (explained in more detail in the second paragraph of 4.1).

## 4.2 Empirical Attack Case Study

To study the empirical effect of $\mathbb{P}_{\mathbf{x}^*}(1)$ on MI positive accuracy, we will observe the positive accuracy of the current state-of-the-art membership inference attack, LiRA (Carlini et al., 2022a), as we vary $\mathbb{P}_{\mathbf{x}^*}(1)$. We also train with different $(\varepsilon, \delta)$-DP guarantees. In the original paper, the attack was not evaluated against DP trained models, so we reimplement the attack ourselves on MNIST, CIFAR10, and SVHN-extended. In particular, we follow Algorithm 1 in Carlini et al. (2022a) but now the training function is DP-SGD, varying $\varepsilon$ with fixed $\delta = 10^{-5}$, and using the hyperparameters and models of Tramèr & Boneh (2020) for training.[3] We train 20 shadow models for each $\varepsilon$ setting (*i.e.* reuse the same shadow models for all the points we test), sampling their training sets uniformly with 50% probability from the original training set. Our target models use the same training function as the shadow models, but we now vary the $\mathbb{P}_{x^*}(1)$ to capture the effectiveness of the attack on target models with different sampling rates.

We evaluate the attack by computing the scores—that Algorithm 1 in Carlini et al. (2022a) outputs—for the first 1000 training points of MNIST, CIFAR10, and SVHN-extended respectively. In each setting, we consider different thresholds for acceptance according to a quantile based approach: we consider accepting the top $i * 2.5\%$ quantile of scores for every $i \in [40]$, and report the best precision (positive accuracy) out of all these different thresholds (*i.e.* select the highest precision threshold).[4]

Figure 3 plots the average positive accuracy (from 5 repeated trials) of the LiRA attack for different $\varepsilon$ as a function of the $\mathbb{P}_{\mathbf{x}^*}(1)$ used for the target model. We further plot the positive accuracy of the baseline attack of always predicting a datapoint is in the training dataset. As we see, the LiRA attack consistently outperforms the baseline, but importantly, **we see that the sampling is the dominant factor dictating**

---

[3]In the case of SVHN-extended, we take the CIFAR10 hyperparameters and model architecture

[4]Due to numerical instability (e.g., the handling of the density value in the tails of the Gaussian), the score is at times a nan. In these cases, our attack predicts the point *was not* a training point.

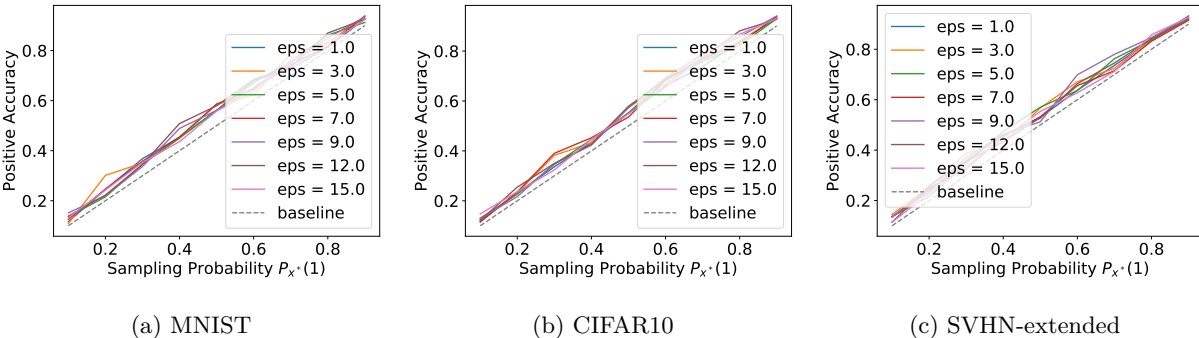

(a) MNIST            (b) CIFAR10            (c) SVHN-extended

Figure 3: Membership inference positive accuracy of the state-of-the-art LiRA attack (Carlini et al., 2022a) on MNIST (left), CIFAR10 (middle), and SVHN extended (right) as a function of the probability $\mathbb{P}_{\mathbf{x}^*}(1)$ of a point being sampled into the training set. The LiRA attack is conducted against models trained with differential privacy. Different curves are obtained by varying the privacy budget $\varepsilon$ used by the differentially private training algorithm (DP-SGD of Abadi et al. (2016)). Observe how sampling is the dominant factor dictating how much the LiRA attack outperforms the baseline attack (*i.e.* predicting a point is always a member).

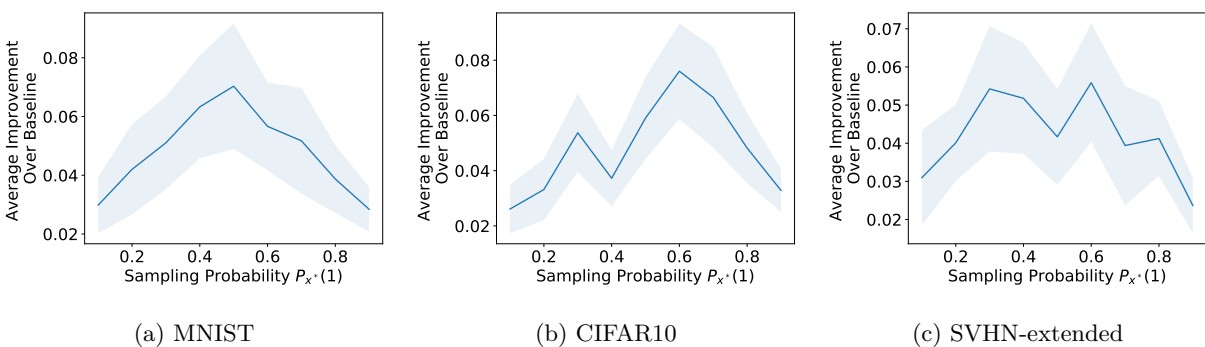

(a) MNIST            (b) CIFAR10            (c) SVHN-extended

Figure 4: Relative improvement of the membership inference positive accuracy for the LiRA attack over the baseline attack on MNIST, CIFAR10, and SVHN-extended. This improvement is plotted as a function of the probability $\mathbb{P}_{\mathbf{x}^*}(1)$ of a point being sampled into the training set. We include the 95% confidence interval (over the 7 values of $\varepsilon$ from Figure 3 and 5 trials). Observe how low sampling probabilities $\mathbb{P}_{\mathbf{x}^*}(1)$ disproportionately mitigate the improvement in membership inference brought by the LiRA attack.

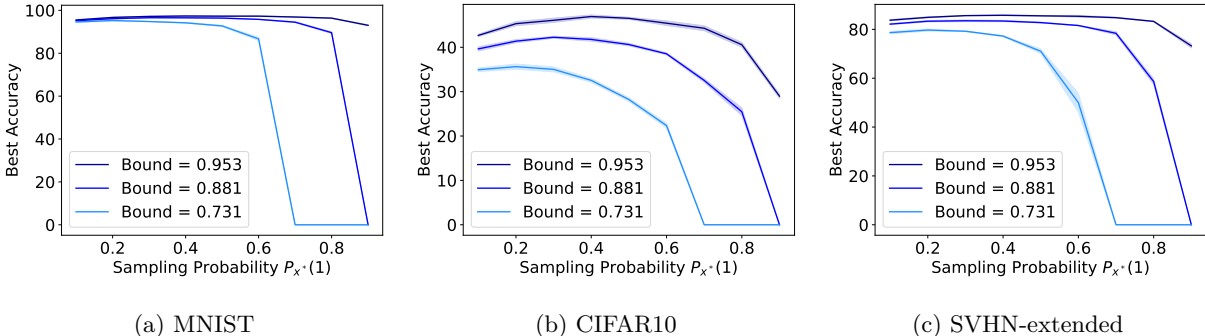

(a) MNIST          (b) CIFAR10          (c) SVHN-extended

Figure 5: Best accuracy for MNIST (left), CIFAR10 (middle) and SVHN-extended (right) as a function of the sampling probability $\mathbb{P}_{\mathbf{x}^*}(1)$. The plots are obtained by fixing a target positive accuracy bound and using Theorem 3.3 to interpolate across the values of $\mathbb{P}_{\mathbf{x}^*}(1)$ and $\varepsilon$ achieving the bound. The bounds are computed assuming $\mathbb{P}(S|\mathbf{x}^*) \geq 1\%$ in Theorem 3.3 We report the best test accuracy from 100 epochs of training. Observe how there is a preference for smaller sampling rates, which is more distinct for smaller fixed positive accuracy bounds. We include the 95% confidence interval over 5 trials.

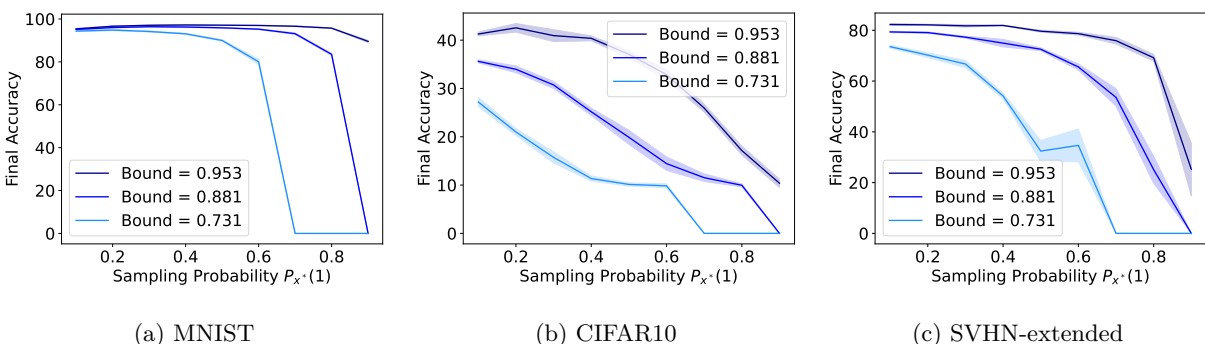

(a) MNIST          (b) CIFAR10          (c) SVHN-extended

Figure 6: Final accuracy for MNIST (left), CIFAR10 (middle) and SVHN-extended (right) as a function of the sampling probability $\mathbb{P}_{\mathbf{x}^*}(1)$. The plots are obtained by fixing a target positive accuracy bound and using Theorem 3.3 to interpolate across the values of $\mathbb{P}_{\mathbf{x}^*}(1)$ and $\varepsilon$ achieving the bound. The bounds are computed assuming $\mathbb{P}(S|\mathbf{x}^*) \geq 1\%$ in Theorem 3.3. We report the final accuracy from 100 epochs of training. Observe how there is a preference for smaller sampling rates, which is more distinct for smaller fixed positive accuracy bounds. We include the 95% confidence interval over 5 trials.

**how much we outperform the baseline.** That is, this state-of-the-art attack can be made less effective by applying our MI subsampling mechanism—as predicted by our bound.

To further illustrate this outweighed effect of sampling over $\varepsilon$; Figure 4 plots the average improvement (over the $\varepsilon$ values from Figure 3 and 5 trials) against the baseline attack. What we see is that both low and high sampling rates see less of an improvement over the baseline than medium sampling rates. In particular, this shows *low sampling disproportionately mitigates membership inference.* This behaviour is consistent with what we expect from our bounds (see Figure 1).

### 4.3 Performance vs. Sampling

We now study the trade-off between sampling to prevent MI and model performance on MNIST, CIFAR10, and SVHN-extended. That is, we fix a target MI bound and vary $\mathbb{P}_{\mathbf{x}^*}(1)$ as we vary $\varepsilon$ according to the formula given in Theorem 3.3 assuming $\mathbb{P}(S|\mathbf{x}^*) \geq 1\%$. That is, bounding only adversaries that predict positively at least 1% of the time when actually training with $x^*$ . This is shown in Figures 5 and 6, where the final accuracy is the accuracy after training for 100 epochs and best accuracy is the highest achieved accuracy during the training run. The fixed target bounds were chosen to be those achieved when setting $P_{\mathbf{x}^*}(1) = 0.5$ and taking $\varepsilon = 3, 2, 1$ with $\delta = 10^{-5}$. In training, we used the models from Tramèr & Boneh (2020), with the training hyperparameters being those reported to perform the best for $\varepsilon = 3$ by Tramèr & Boneh (2020).

We observe that MNIST often finds an optimal accuracy with $\mathbb{P}_{\mathbf{x}^*}(1) = 0.5$, though for the tightest bound we observe there is a preference for smaller $\mathbb{P}_{\mathbf{x}^*}(1)$. CIFAR10 suffers more disproportionately from having smaller epsilons than having smaller datasets, tending to have an optimal accuracy with $\mathbb{P}_{\mathbf{x}^*}(1) = 0.3$. This is even more apparent when looking at the final accuracy for CIFAR10, where $\mathbb{P}_{\mathbf{x}^*}(1) = 0.1$ often performs best. We conclude that CIFAR10 performance *benefits from low sampling (with some fixed target MI bound).* SVHN-extended follows similar trends as CIFAR10, with a stronger preference of low $\mathbb{P}_{\mathbf{x}^*}(1)$ when considering smaller fixed MI bounds (e.g., the bound when using $\varepsilon = 1$ with $\mathbb{P}_{\mathbf{x}^*}(1) = 0.5$).

So to conclude, for MNIST, CIFAR10, and SVHN-extended we empirically observe that lower sampling, i.e., training with less data, is beneficial for performance.

## 5 Why Bound Membership Inference?

So far we have motivated the usefulness of studying membership inference by the fact membership inference is currently the most practical attack against DNNs. In this section, we further connect the pursuit of bounding membership inference to other privacy attacks and topics of general interest to the ML community.

### 5.1 Importance to Data Deletion

The ability to decrease MI positive accuracy, *i.e.* the ability for an arbitrator to attribute a data point to a model, has consequences for machine unlearning and data deletion. Cao & Yang (2015) were the first to describe a setting where it is important for the model to be able to "forget" certain training points. The authors focused on the cases where there exist efficient analytic solutions to this problem. The topic of machine unlearning was then extended to DNNs by Bourtoule et al. (2019) with the definition that a model has unlearned a data point if after the unlearning, the distribution of models returned is identical to the one that would result from not training with the data point at all. This definition was also stated earlier by Ginart et al. (2019) for other classes of machine learning models.

Given that unlearning is interested in removing the impact a data point had on the model, further work employed MI accuracy on the data point to be unlearned as a metric for how well the model had unlearned it after using some proposed unlearning method (Baumhauer et al., 2020; Graves et al., 2020; Golatkar et al., 2020b;a). Yet, empirical estimates on the membership status of a datapoint are subjective to the concrete MI attacks employed. Indeed it may be possible that there exists a stronger attack. The issue of empirical MI for measuring unlearning was recently illustrated by Thudi et al. (2021) and Carlini et al. (2022b) where they highlighted undesirable disconnects between it and other measures of unlearning.

**Applying our result to unlearning.** Analytic bounds to MI attacks, on the other hand, resolve the subjectivity issue of a specific MI attack as a metric for unlearning as they bound the success of any adversary. In particular one could give the following definition of an unlearning guarantee from a formal MI positive accuracy bound:

**Definition 5.1** ($B$-MI Unlearning Guarantee). An algorithm is $B$-MI unlearnt for $\mathbf{x}^*$ if $\mathbb{P}(\mathbf{x}^* \notin D|S) \geq B$, *i.e.* the probability of $\mathbf{x}^*$ not being in the training dataset is greater than $B$.

Therefore, our result bounding positive MI accuracy has direct consequences on the field of machine unlearning. In particular if $\mathbb{P}(\mathbf{x}^* \in D|S)$ is sufficiently low, that is the likelihood of $S$ coming from $\mathbf{x}^*$ is low, then an entity could claim that they do not need to delete the users data since their model is most likely independent of that data point as it most likely came from a model without it: *i.e.* leading to plausible deniability. This is similar to the logic presented by Sekhari et al. (2021) where unlearning is presented probablistically in terms of $(\varepsilon, \delta)$-unlearning.

We also observe an analogous result to Sekhari et al. (2021) where we can only handle a maximum number of deletion requests before no longer having sufficiently low probability. To be exact, let us say we do not need to undergo any unlearning process given a set of data deletion request $\hat{\mathbf{x}}^*$ if $\mathbb{P}(\hat{\mathbf{x}}^* \notin D|S) \geq B$ for some $B$. Now, for simplicity, assume the probability of drawing all points into the datasets are the same and, so that for all $\mathbf{x}_i^*$ we have the same bound given by Theorem 3.3, that is $\mathbb{P}(\mathbf{x}_i^* \notin D|S) > L$ for some $L \leq 1$.[5] Then we have $\mathbb{P}(\hat{\mathbf{x}}^* \notin D|S) \geq L^m$ as if $\hat{\mathbf{x}}^* = \{\mathbf{x}_1^*, \mathbf{x}_2^*, \cdots, \mathbf{x}_m^*\}$ then $\mathbb{P}(\hat{\mathbf{x}}^* \notin D|S) = \mathbb{P}(\mathbf{x}_1^* \notin D|S) \cdots \mathbb{P}(\mathbf{x}_m^* \notin D|S, \{\mathbf{x}_1^*, \mathbf{x}_2^*, \cdots, \mathbf{x}_{m-1}^*\} \notin D)$ but this conditioning on the other datapoints not being sampled does not change our lower-bound.[6] Hence, an entity does not need to unlearn if $L^m \geq B$, *i.e.* if $m \leq \frac{\ln 1/B}{\ln 1/L} = \frac{\ln B}{\ln L}$. This gives a bound on how many deletion requests the entity can avoid in terms of the lower bound on negative accuracy implied by the upper-bound on positive accuracy given in Theorem 3.3. In particular, if $\{\mathbf{x}_1 \cdots \mathbf{x}_N\}$ is the larger set of data points an entity is sampling from, and $\mathbb{P}_{\mathbf{x}}(1) = c/N \ \forall \mathbf{x} \in \{\mathbf{x}_1 \cdots \mathbf{x}_N\}$, then the lower bound (taking $\delta = 0$) is $\left(1 + \frac{e^{-\varepsilon}\frac{c}{N}}{1-\frac{c}{N}}\right)^{-1}$.

**Corollary 5.2** ($B$-MI Unlearning Capacity). *If $\{\mathbf{x}_1 \cdots \mathbf{x}_N\}$ is the larger set of data points an entity is sampling from, and $\mathbb{P}_{\mathbf{x}}(1) = c/N \ \forall \mathbf{x} \in \{\mathbf{x}_1 \cdots \mathbf{x}_N\}$, then one can delete $m \leq \frac{\ln B}{\ln L(c)}$ with $L(c) = \left(1 + \frac{e^{-\varepsilon}\frac{c}{N}}{1-\frac{c}{N}}\right)^{-1}$ and satisfy $B$-MI unlearning.*

Sekhari et al. (2021) showed that with typical DP the deletion requests grow linearly with the size of the training (in the above case $c$ represents the expected training set size). We thus compare a linear line w.r.t $c$ to $\frac{\ln B}{\ln L(c)}$ (given by Corollary 5.2) in Figure 7 to observe their respective magnitude: we fix $B = 0.8$, $N = 10000$ and $\varepsilon = 1$ as we are interested in asymptotics. We observe that our deletion capacity is significantly higher for low expected training dataset sizes and is marginally lower than a linear trend for larger training set sizes.

## 5.2 Applying to Data Reconstruction

An interesting question is how training with DP impacts the effectiveness of data reconstruction attacks. The goal of a data reconstruction attack is, given a model (and/or the sequence of models used to obtain it), to have an algorithm that outputs a training point: more realistically, a point close to a training point. In this sense, the attack "reconstructed" a training datapoint. An immediate consequence of our MI positive accuracy bounds is a statement that data reconstruction attacks are not trustworthy; even when they reconstruct an image close to a possible datapoint, the probability of the datapoint actually being a training datapoint is bounded.

**Corollary 5.3** (Data Reconstruction not Trustworthy).[7] *Consider a deterministic[7] "data reconstruction" algorithm $R$ that takes a model (or the training sequence of models) $M$ and outputs a datapoint. We say it*

---

[5]This $L$ is given by 1 minus the upper bound on positive accuracy

[6]It changes the larger set of possible data, but the bound is agnostic to what this set is and only depends on the individual sampling probability

[7]One could consider random algorithms, but fixing the random seed the bound will apply. As the bound applies for every random seed, it also applies in expectation.

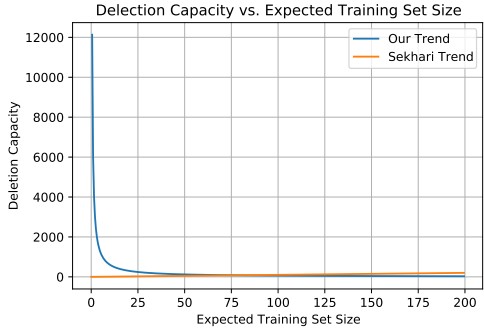

Figure 7: Comparing our deletion capacity trend to the trend Sekhari et al. (2021) describes. In particular, our deletion capacity grows as the expected training set size decreases.

*"reconstructs" a training datapoint out of the set of possible datapoints $\mathbb{D} \coloneqq \{\mathbf{x}_1, \cdots, \mathbf{x}_N\}$ if for some metric $d$, $d(R(M), \mathbf{x}_i) \leq \varepsilon$ for some $\varepsilon \geq 0$. For any $\mathbf{x}_i$, consider the set $S \coloneqq \{M : d(R(M), \mathbf{x}_i) \leq \varepsilon\}$ of models it claims to have trained on $\mathbf{x}_i$. By Theorem 3.3, when training with $(\varepsilon, \delta)$-DP, the probability $\mathbf{x}_i$ was actually used to train models in $S$ is $\mathbb{P}(\mathbf{x}^*|S) \leq (1 + \frac{e^{-\varepsilon}\mathbb{P}_{\mathbf{x}^*}(0)}{\mathbb{P}_{\mathbf{x}^*}(1)} - \frac{\delta e^{-\varepsilon}\mathbb{P}_{\mathbf{x}^*}(0)}{\mathbb{P}(S|\mathbf{x}^*)})^{-1}$.*

*Proof.* Just an application of Theorem 3.3 □

However, we believe a more interesting question is if, considering a specific class of reconstruction algorithms $R$, we can lower bound the reconstruction error (with high probability). We believe this should be possible, but currently are not sure what an appropriate class of $R$ is. A naive idea is that $d(R(M), \mathbf{x}) \leq \varepsilon$ iff $\mathbf{x} \in D$, that is, the algorithm can only reconstruct datapoints used in the training set. However, the above corollary states no such attacks can exist when training with $\varepsilon$-DP. So it stands this characterization might be too powerful. We leave finding fruitful classes of reconstruction attacks to study for future work.

### 5.3 We can Bound Generalization Error

In this subsection we highlight a connection between bounds on MI adversaries and generalization error. Yeom et al. (2018) studied MI adversaries in the $\mathbb{P}_{\mathbf{x}^*}(1) = 0.5$ setting, *i.e.* 50% probability of being sampled, that took advantage of a generalization gap $R_g$ between training and non-training datapoints. Specifically, they described an adversary whose accuracy is $(R_g/B+1)/2$, where $B$ is an upper-bound on the loss (Theorem 2 in Yeom et al. (2018)). However, note that this accuracy is upper and lower-bounded by Theorem 3.3 as the theorem bounds the accuracy of all adversaries when $\mathbb{P}_{\mathbf{x}^*}(1) = 0.5$. The lower-bound is because the upper-bound on negative accuracy gives a lower-bound on positive accuracy, and similarly the upper-bound on positive accuracy gives a lower-bound on negative accuracy. Thus we have the following corollary.

**Corollary 5.4** (MI bounds Generalization Error). *For loss function upper-bounded by $B$ and an $\varepsilon$-DP training algorithm, we have $(1 + e^{\varepsilon})^{-1} \leq (R_g/B + 1)/2 \leq (1 + e^{-\varepsilon})^{-1}$ where $R_g$ is the generalization gap (Definition 3 in Yeom et al. (2018), where their $D$ is our larger set, and their $S$ is the specific training set sampled from the larger set).*

This tells us how well the $\mathbb{P}_{\mathbf{x}^*}(1) = 0.5$ sampled dataset generalizes to the entire larger set. We believe this general procedure of using specific adversaries with accuracies dependent on values of interest (*e.g.* the generalization gap here) and applying more general bounds to get a range for those values of interest is something future work can expand on.

## 6 Conclusion

The current paradigm for boosting performance when obtaining private models is to train on more, often public, data. In contrast to this, we showed how training on less data can be beneficial for the privacy vs.

performance trade-off. This was done by first obtaining a new membership inference bound that more tightly captures the effect of small training set sampling. In Appendix B, we give more details on how our bound relates to prior bounds. This allowed us to describe a new privacy amplification mechanism that a defender can reasonably take advantage of, and we showed that the disparate effect of this mechanism can be seen empirically in the current state-of-the-art membership inference attack. Studying the effect of this sampling mechanism on performance, we showed that what we gained in performance by being able to use weaker DP guarantees outweighs what we lost by training on less data. Ultimately, smaller training sampling rates can lead to performance gains while retaining specific privacy guarantees.

## Broader Impact

Our results show that under specific threat models and metrics of privacy leakage, we can have alternative ways of protecting privacy when using machine learning complementary to differential privacy, leading to new "sweet spots" for model performance. In particular, we have shown that when an entity wants to claim deniability of training on a datapoint, such as for unlearning, modifying their sampling rate of training points is one method to do so which allows them to train with weaker DP guaranatees. We believe this result motivates further consideration of the different real-world threat models and the methods to ensure "privacy" in those threat models, such as the role of priors (i.e., sampling rates) and what the attack can use (i.e., deterministic given the models returned in training).

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

# A   Further Discussion on The Bound

## A.1   Positive Accuracy is not always bounded

Here we present why the failure mode in $\mathbb{P}(S|x^*)$ is necessary.

Consider a set of two (scalar) points $\{x_1, x_2\}$ which are drawn into the training set $D$ with $\mathbb{P}_{x_1}(1) = 1$ and $\mathbb{P}_{x_2}(1) = 0.5$; that is $x_1$ is always in the training set, and $x_2$ has a 50% chance of being in the training set. Let model $M$ be a single dimensional logistic regression without bias defined as $M(x) = wx$ initialized such that for cross-entropy loss $L$, $\nabla L|_{\mathbf{x}_1} \approx 0$ (*i.e.* set $x_1 = \{(10^6, 1)\}$ and the weights $w = 10^6$, so that $M(x) = 10^6 x$ and thus the softmax output of the model is approximately 1 on $x_1$ and thus gradient is approximately 0). Conversely set $x_2$ such that the gradient on it is less than $-1$ (*i.e.* for the above setting set $x_2 = \{(10^6, 0)\}$).

Now, train the model to $(1, 10^{-5})$-DP following Abadi et al. (2016) with $\eta = 1$, sampling rate of 100%, a maximum gradient norm of 1, for one step. Note these are just parameters which we are allowed to change under the DP analysis, and we found the noise we would need for $(1, 10^{-5})$-DP is 4.0412. Then consider

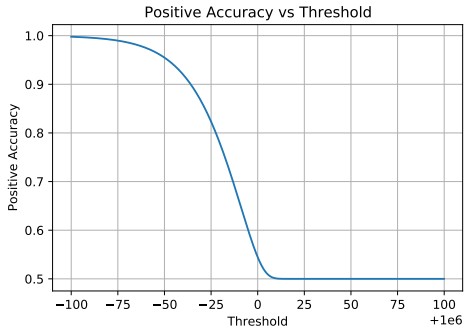

(a) Positive accuracy as a function of the threshold

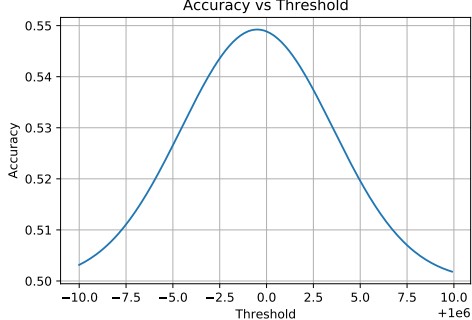

(b) Accuracy as a function of the threshold

Figure 8: Impact of threshold on positive accuracy and accuracy.

running a MI attack on the above setup where if for some threshold $\alpha$ the final weights $\mathbf{W}$ are s.t if $\mathbf{W} \leq \alpha$ one classifies those weights as having come from the dataset with $x_2$, otherwise not. Do note that here we use $\mathbf{W}$ for the final weights as opposed to $w$ to emphasize that we are now talking about a random variable. The intuition for this attack is that if the dataset only contain $x_1$ then the weights do not change, but if the dataset contains $x_2$ we know the resulting gradient is negative (by construction) and thus decreases the weights (before noise).

By the earlier setting on how training datasets are constructed note that $D = \{x_1\}$ or $D = \{x_1, x_2\}$, and we will denote these $D_1$ and $D_2$ respectively. Note that if $M$ trained on $D_1$ following the suggested data points and initial weights, we have the distribution of final weight $\mathbf{W}_{D_1} = N(10^6, \sigma) = N(10^6, 4.0412)$ where $\sigma$ denotes the noise needed for $(1, 10^{-5})$-DP as stated earlier. Similarly $\mathbf{W}_{D_2} = N(10^6 - 1, 4.0412)$, since the maximum gradient norm is set to 1.

For the above MI attack we can then explicitly compute the positive accuracy as a function of $\alpha$ by:

$$\mathbb{P}(D_2|\mathbf{W} \leq \alpha)$$

$$= \frac{\mathbb{P}(\mathbf{W}_{D_2} \leq \alpha) * \mathbb{P}(D_2)}{\mathbb{P}(\mathbf{W}_{D_2} \leq \alpha) * \mathbb{P}(D_2) + \mathbb{P}(\mathbf{W}_{D_1} \leq \alpha) * \mathbb{P}(D_1)}$$

$$= \frac{\phi(\mathbf{W}_{D_2}, \alpha) * 0.5}{\phi(\mathbf{W}_{D_1}, \alpha) * 0.5 + \phi(\mathbf{W}_{D_2}, \alpha) * 0.5}.$$

Note $\phi(\mathbf{W}, \alpha)$ is the (Gaussian) cumulative function of random variable $\mathbf{W}$ upto $\alpha$. We plot this in Figure 8a, and note how it is not bounded by anything less than 1 and goes to 1 as the threshold $\alpha$ decreases (*i.e.* $\forall m \in [0, 1)$ $\exists \alpha$ s.t $S = (-\infty, \alpha]$ yields positive accuracy greater than $m$).

# B   Previous Bounds Compared to Our Bounds

In our work we are interested in providing bounds on MI that better describe non-DP influences on the effectiveness of an attack. However, to situate our work, we first describe the context prior work derived MI bounds in. Two of the main bounds (Yeom et al., 2018; Erlingsson et al., 2019) focused on an experimental setup first introduced by Yeom et al. (2018). In this setting, an adversary $f$ is given a datapoint $\mathbf{x}^*$ that is 50% likely to have been used to train a model or not. The adversary then either predicts 1 if they think it was used, or 0 otherwise. Let $b = 1$ or 0 indicate if the datapoint was or was not used for training, respectively; we say the adversary was correct if their prediction matches $b$. We then define the adversary's advantage as improvement in accuracy over the 50% baseline of random guessing, or more specifically $2(A(f) - 0.5)$ where $A(f)$ is the accuracy of $f$.

For such an adversary operating in a setting where data is equally likely to be included or not in the training dataset, Yeom et al. (2018) showed that they could bound the advantage of the adversary by $e^\varepsilon - 1$ when training with $\varepsilon$-DP. Their proof used the fact that the true positive rate (TPR) and false positive rate (FPR) of their adversary could be represented as expectations over the different data points in a dataset, and from that introduced the DP condition to obtain their MI bound.

Erlingsson et al. (2019) improved on the bound developed by Yeom et al. (2018) for an adversary operating under the same condition by utilizing a proposition given by Hall et al. (2013) on the relation between TPR and FPR for an $(\varepsilon, \delta)$-DP function. Based on these insights, Erlingsson et al. (2019) bounded the membership advantage by $1 - e^{-\varepsilon} + \delta e^{-\varepsilon}$. Finally, improving on Erlingsson et al. (2019) by using propositions given by Kairouz et al. (2015), Humphries et al. (2020) bounded the membership advantage by $\frac{e^\varepsilon - 1 + 2\delta}{e^\varepsilon + 1}$. This translates to an accuracy bound of $\frac{2e^\varepsilon + 2\delta}{2(e^\varepsilon + 1)}$ which is the tightest MI accuracy bound we are aware of.

Other work has considered variable sampling rates. For $\varepsilon$-DP, Sablayrolles et al. (2019) bounded the probability of a datapoint $\mathbf{x}^*$ being used in the training set of a model (*i.e.* , $\mathbb{P}(\mathbf{x}^* \in D_{train}|S)$ which we will refer to as "positive accuracy") by $\mathbb{P}_{\mathbf{x}^*}(1) + \frac{\varepsilon}{4}$ where $\mathbb{P}_{\mathbf{x}^*}(1)$ is the probability of the datapoint being sampled into the training dataset. [8]

We now discuss how the bounds obtained in this line of work compare to our bound, obtained in Theorem 3.3. For simplicity, let us take $\delta = 0$. Note that the setting described in Section 3.2 is equivalent to the MI experiment defined by Yeom et al. (2018) when $\mathbb{P}_{\mathbf{x}_i}(1) = 0.5 \; \forall \mathbf{x}_i \in \{\mathbf{x}_1, \cdots, \mathbf{x}_N\}$. Moreover, note the positive accuracy bound in Theorem 3.3 also bounds MI accuracy for $\mathbb{P}_{\mathbf{x}}(1) = 0.5$. This is as setting $\delta = 0$, Theorem 3.3 gives a bound for $\epsilon$-DP algorithms. Note then that by taking $\mathbb{P}_{\mathbf{x}^*}(1) = \mathbb{P}_{\mathbf{x}^*}(0) = 0.5$, the positive and negative accuracy bounds are equal in Theorem 3.3. Therefore, as an adversary must output either a positive or a negative prediction, we have MI accuracy (not positive accuracy or negative accuracy) is bounded by Theorem 3.3 when $\mathbb{P}_{\mathbf{x}^*}(1) = 0.5$.

Note the bound we obtain is $(1 + e^{-\varepsilon})$ which matches the MI accuracy bound given by Humphries et al. (2020), the previous tightest bound on MI accuracy (see Figure 9). One of the contributions of our proof is it explicitly demonstrates where the looseness in this bound comes from, *i.e.* looseness of the DP inequality over all $(D, D')$ pairs.

Do note our positive accuracy bounds extend to the setting with a biased sampler. We compare our bound with the positive accuracy bound derived by Sablayrolles et al. (2019) in Figure 10 which demonstrates that our bound is indeed tighter.

Concurrent work by Mahloujifar et al. (2022) improved the bound on positive accuracy by Sablayrolles et al. (2019) (also called precision or positive predictive value "when referring specifically to an adversary") to $\frac{1}{1 + e^{-\varepsilon}}$ in the case of $\mathbb{P}_{\mathbf{x}^*}(1) = 0.5$ and all other data is fixed. Note eur bound on positive accuracy generalizes to a more general setting than the one Mahloujifar et al. (2022) is limited to.

---

[8]We note that Jayaraman et al. (2020) bounded the positive predictive value (equivalent to precision and positive accuracy) of an attacker on a model trained with $(\varepsilon, \delta)$-DP when the FPR is fixed. Note, that both our work and the previously mentioned bounds are independent of FPR. In particular Erlingsson et al. (2019) followed a similar technique to Jayaraman et al. (2020), but were able to drop the FPR term using a proposition relating TPR to FPR (Hall et al., 2013).

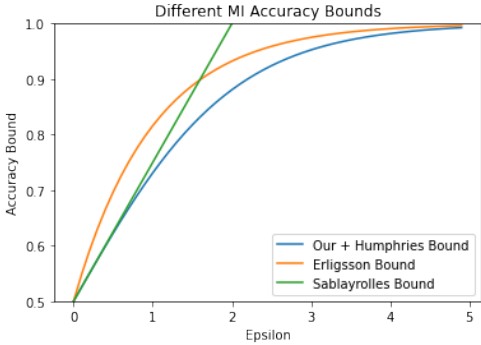

Figure 9: Comparing the upper bound to MI performance we achieved matching Humphries et al. (2020) to that given by Erlingsson et al. (2019) and Sablayrolles et al. (2019) (note $\mathbb{P}_{\mathbf{x}^*}(1) = 0.5$ here). In particular, note we are tighter for all $\varepsilon$.

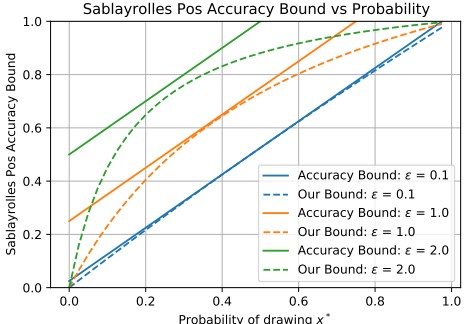

Figure 10: Sablayrolles et al. (2019) upper bound on MI positive accuracy as a function of $\mathbb{P}_{\mathbf{x}^*}(1)$ compared to our bound. Note that we are still tighter for all probabilities.

In particular, we now describe how our analysis yields a more relevant (and stronger) approach to bounding failure rates when using $(\varepsilon, \delta)$-DP than the $(1 - \delta)$ approach used by Mahloujifar et al. (2022). Following Eq. 15 in Mahloujifar et al. (2022) which states the $\varepsilon$-DP conditions holds for any fixed $(D, D')$ pair with probability $(1 - \delta)$, the bounds in Theorem 3.3 (setting $\delta = 0$) trivially apply to $(\varepsilon, \delta)$-DP with probability $(1 - (\delta))^N$ where $N$ is the number of $(D, D')$ pairs appearing from the random sampling. Note this $N$ could be quite large, *i.e.* if $n$ is the number of datapoints in the larger set, $N = \sum_{i=1}^{n-1} \binom{n-1}{j}$ if every point is being randomly sampled. The scenario studied by Mahloujifar et al. (2022) bypasses the $N$ dependence by only considering one $(D, D')$ pair, *i.e.* all the other points in the dataset are fixed and only the datapoint in question is being randomly sampled. This assumption made by Mahloujifar et al. (2022) is not applicable in practice as we do not know which datapoints the adversary is going to test: hence, we would have to resort to randomly sampling all of them. Instead, our failure analysis in Section 3.3 (particularly Theorem 3.3) conveniently bypasses this problem by telling us low probability states are what lead to failures (removing the $N$ dependence in the failure rate).

As a last remark of how the work of Mahloujifar et al. (2022) can be viewed in the context of our result, recall our bound is tight upto the looseness of the DP inequalities holding on all adjacent datasets. Hence, any future improvement would have to go beyond the DP condition and incorporate the looseness of the DP guarantee on different datasets. This means that our bound is optimal when no such knowledge is provided. Mahloujifar et al. (2022) does just that by studying MI accuracy bounds given by a privacy mechanism, beyond just the $(\varepsilon, \delta)$-DP guarantees it yields. However it should be noted that MI accuracy, which is what Mahloujifar et al. (2022) primarily studies, and MI positive accuracy (what we study) have divergent analyses in the case of $(\varepsilon, \delta)$-DP due to failure mode described above (which is not present for accuracy).

## C    Additional Figures

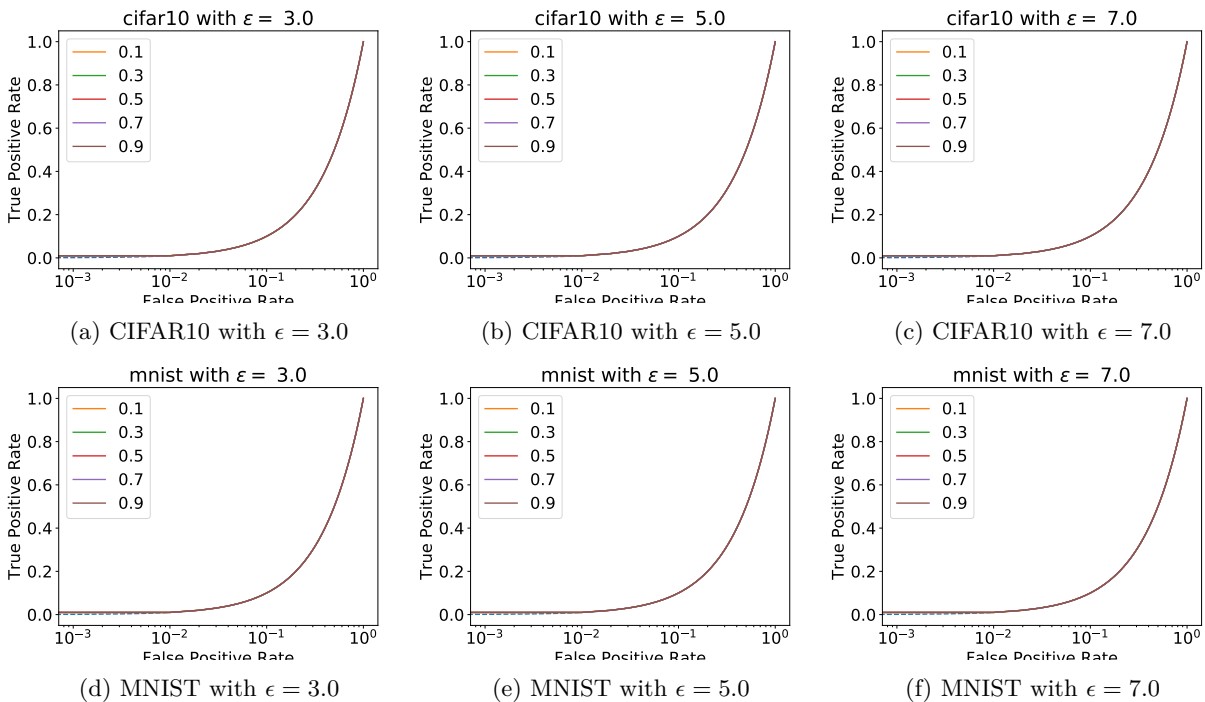

Figure 11: Viewing privacy as hypothesis testing Wasserman & Zhou (2010), we present TPR vs. FPR for LiRA attack against MNIST and CIFAR10 for $\epsilon = 3.0, 5.0, 7.0$ and $\delta = 10^{-5}$, with the dashed line representing $TPR = FPR$. For each setting, we plotted the attack against target models with sampling rates $P_{\mathbf{x}^*} = 0.1, 0.3, 0.5, 0.7, 0.9$. The results are the average over 5 trials with a 95% confidence interval. We see marginal strength at low false positive rates.

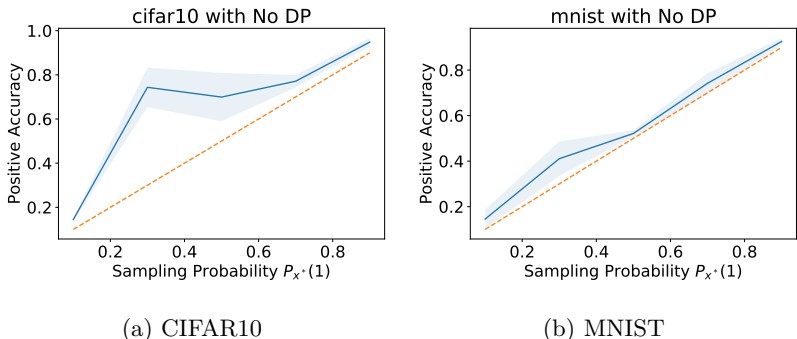

Figure 12: Positive accuracy of the LiRA attack against MNIST and CIFAR10 when training with no privacy guarantees (normal SGD). For each setting, we plotted the attack against target models with sampling rates $P_{\mathbf{x}^*} = 0.1, 0.3, 0.5, 0.7, 0.9$. The results are the average over 5 trials with a 95% confidence interval, and the baseline (always predicting positive) is given by the dashed line. We observe that the improvement over the baseline depends on the sampling rate $P_{\mathbf{x}^*}$.

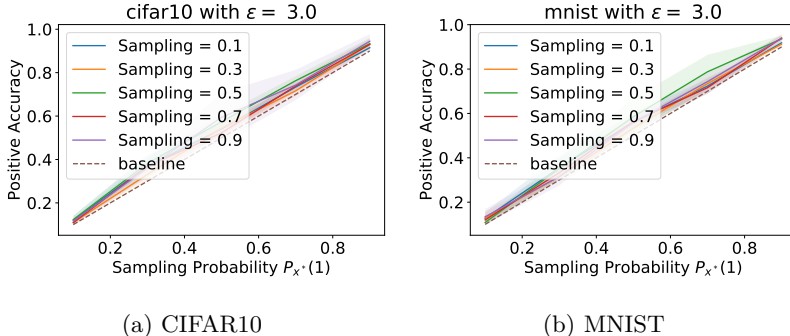

(a) CIFAR10                                              (b) MNIST

Figure 13: Positive accuracy of the LiRA attack against MNIST and CIFAR10 when training with ($\epsilon = 3.0, \delta = 10^{-5}$)-DP. For each setting, we plotted the attack against target models with sampling rates $P_{\mathbf{x}^*} = 0.1, 0.3, 0.5, 0.7, 0.9$ when using shadow models trained with varying sampling rates. The results are the average over 5 trials with a 95% confidence interval, and the baseline (always predicting positive) is given by the dashed line. We observe that there is no significant difference between the sampling rates, though $P_{\mathbf{x}^*}(1) = 0.5$ is on average best.

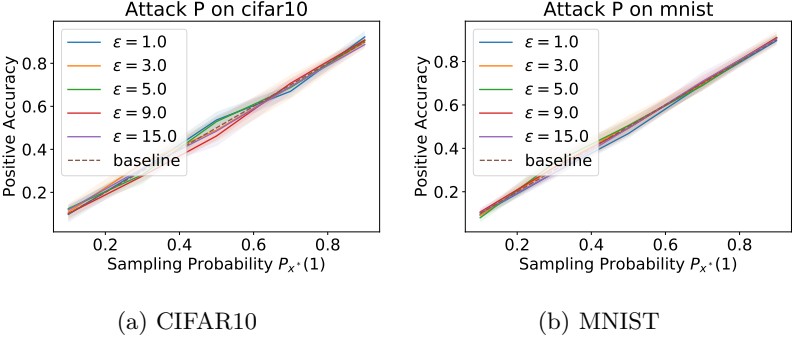

(a) CIFAR10                                              (b) MNIST

Figure 14: Positive accuracy of Attack P in Ye et al. (2022) against MNIST and CIFAR10 when training with $\epsilon = 1.0, 3.0, 5.0, 7.0, 9.0, 15.0$ and $\delta = 10^{-5}$). We implemented Attack P by using 1000 test points (non training points) to set the threshold on the loss to ensure a FPR of 0.1. For each privacy setting, we plotted the attack against target models with sampling rates $P_{\mathbf{x}^*} = 0.1, 0.3, 0.5, 0.7, 0.9$. The results are the average over 5 trials with a 95% confidence interval, and the baseline (always predicting positive) is given by the dashed line. We observe no significant improvement over the baseline with respect to $P_{\mathbf{x}^*}$.

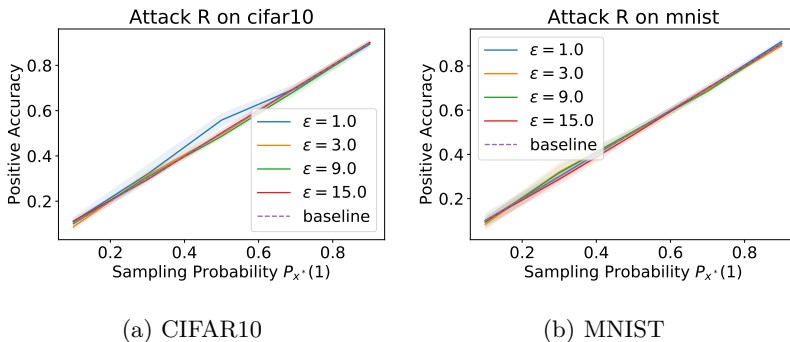

(a) CIFAR10          (b) MNIST

Figure 15: Positive accuracy of Attack R in Ye et al. (2022) against MNIST and CIFAR10 when training with $\epsilon = 1.0, 3.0, 5.0, 7.0, 9.0, 15.0$ and $\delta = 10^{-5}$). We implemented Attack R by training 20 shadow models sampling training datapoints with probability 50%, and computing the loss among the shadow models that did sample a datapoint to pick a datapoint dependent threshold for the loos to ensure a FPR of 0.1. For each privacy setting, we plotted the attack against target models with sampling rates $P_{\mathbf{x}^*} = 0.1, 0.3, 0.5, 0.7, 0.9$. The results are the average over 5 trials with a 95% confidence interval, and the baseline (always predicting positive) is given by the dashed line. We observe no significant improvement over the baseline with respect to $P_{\mathbf{x}^*}$.

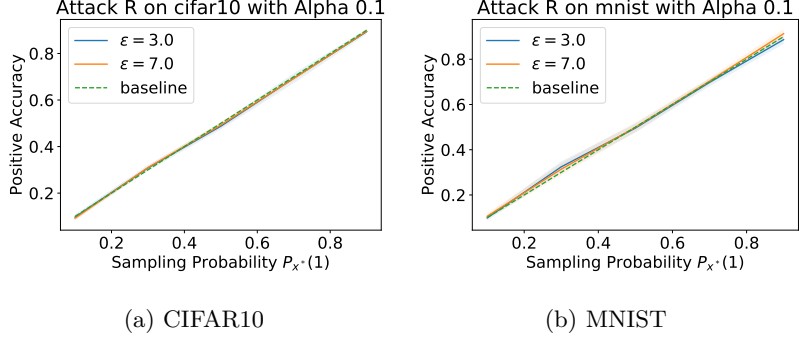

Figure 16: Positive accuracy of Attack R in Ye et al. (2022) against MNIST and CIFAR10 following the setup of Figure 15, but now with different FPR constraints among the shadow models to pick the threshold. We again see now significant improvement over the baseline.

(a) CIFAR10

(b) MNIST

Figure 17: Positive accuracy of Attack R in Ye et al. (2022) against MNIST and CIFAR10 when training with $\epsilon = 3.0, 7.0$ and $\delta = 10^{-5}$), following the setup of Figure 15 except the shadow models sample data at the same rate as the target model. We observe no significant improvement over the baseline with respect to $P_{\mathbf{x}^*}$.

