# OpenReview forum: "From Differential Privacy to Bounds on Membership Inference: Less can be More"
_TMLR — Accepted by TMLR_

### Review · Reviewer_yek7 · 2023-10-31

**Summary Of Contributions:**

- This article presents an upper bound on the MI accuracy when the model is DP-trained.
- This article discovers that sabsampling the training set can amplify the privacy and reduce MI accuracies.
- This article draws connections between the proposed bound and a few other important concepts including data deletion, reconstruction, and generalization bound.

**Audience:**

Yes

**Broader Impact Concerns:**

It would be nice to have a short discussion on what people should take care of when they use MI attacks in practice (for example, to obey legal restrictions or to go to court) given the discovery of this article.

**Claims And Evidence:**

Yes

**Requested Changes:**

- Refine the writing in Section 4.1 and make it easier to understand.
- Make clear definition of what "model utility" means.
- Experiment with more recent MI attacks and conclude if the same phenomenon can be observed.

**Strengths And Weaknesses:**

Strengths:
- The paper is well-motivated, aiming at a very important question in ML privacy research today.
- The proposed bound reveals the relationship between DP and MI accuracy, which is novel and significant to understand the connection between these two concepts.
- The discovery that subsampling can amplify privacy is very interesting and surprising finding, and experiments on various datasets have validated this idea for the LiRA MI attack.
- There is a very excellent, theoretically grounded connection to deletion, reconstruction, and generalization in Section 5.

 Weaknesses:
- The second paragraph of Section 4.1 is a bit confusing to me. I hope the authors could explain the approach a bit further in this paragraph. What is the relationship between $D$ and $D_{\mathrm{train}}$ in this context? Is $T$ a independent or dependent with $x$?
- It is unclear what "model utility" means in this article.
- There lack experiments on more recent SOTA MI attacks such as "Enhanced membership inference attacks against machine learning models" by J. Ye et al, etc.

---

> ### Author Response · Authors · 2023-12-21
> **Response to Reviewer yek7**
>
> > *The second paragraph of Section 4.1 is a bit confusing to me. I hope the authors could explain the approach a bit further in this paragraph. What is the relationship between D and D_train in this context? Is T a independent or dependent with x?*
>
> We assume $D_{train} \subset D$ where each $x \in D_{train}$ is sampled from $D$ independently with probability $T$. Note that the $x \in D$ were sampled with unknown probability, and are possibly not i.i.d. We have also added the following ``real-world” example of the procedure we described:
>
> "To give a real-world example, assume $D$ is obtained by scraping wikipedia articles, following some protocol that may be deterministic. In this case, without exact knowledge of the sampling probabilities of points, and a guarantee they were independent, we cannot use our bound. However, given $D$, now consider sampling from it specific sentences (i.e., inputs) independently, where each sentence has probability $T$  of being included into the new dataset $D_{train}$. We then train our model using DP with $D_{train}$. We then can apply Theorem 3.3 to bound the positive accuracy any adversary (that makes deterministic decisions given the models returned during training) can have by setting $\mathbb{P}_{x^*}(1) =  T$ in the Theorem (explained in more detail in the second paragraph of 4.1)."
>
> > *It is unclear what "model utility" means in this article.*
>
> We mean model accuracy on test distributions, i.e., performance. We have replaced the use of "utility”  with "performance” throughout the paper.
>
> > *There lack experiments on more recent SOTA MI attacks such as "Enhanced membership inference attacks against machine learning models" by J. Ye et al, etc.*
>
> As suggested, we have now included results for Attack P and Attack R in the cited paper. Note we chose these attacks because they are representative: in the cited paper, they report two clusters in performance, one which contains Attack P and the other which contains Attack R.
>
> We implemented Attack P by using 1000 test points (non training points) to set the threshold on the loss to ensure a desired FPR of the attack. Similarly, for Attack R we implemented it by first sampling training datapoints independently with probability 50% and training shadow models on those sets. We then took the shadow models that did not sample the datapoint to calibrate a threshold on the loss to ensure a FPR for the attack. Note this is done by finding the bottom $\alpha$ percentile of losses for those shadow models, and taking that as the threshold to get an $\alpha$ FPR.
>
> As shown in Figures 14 and 15, for these attacks we do not see a consistent improvement over the baseline, instead we observe that they have no predictive power against the DP trained models. In particular, for Attack R we considered further ablations along the FPR used to set the threshold (Figure 16) and the sampling rate used for the shadow models (Figure 17). We saw no consistent improvement in these ablations.
>
> We wish to point out that, to the best of our knowledge, most privacy auditing attacks use LiRA, and our results suggest that against a privately trained model LiRA performs better (in terms of positive accuracy) than Attack P and Attack R from "Enhanced membership inference attacks against machine learning models". In other words, our bounds depend on a DP guarantee, and to best examine the bound one needs an attack that gets close to the DP guarantee of the models.
>
> Note that this is a limitation of the privacy attacks in these dataset settings, rather than a limitation of our bound. In particular, our main experimental results on the sweet spot of sampling rates for model performance uses our theoretical bound against all adversaries, and it is tight given the DP guarantee is tight for all possible datasets under sampling. Our results for specific MI attacks is a study of how close empirical attacks are to this theoretical bound, and how stronger privacy attacks exhibit a dependence on sampling rate (which one can leverage to bound positive accuracy).
>
> > *It would be nice to have a short discussion on what people should take care of when they use MI attacks in practice (for example, to obey legal restrictions or to go to court) given the discovery of this article.*
>
> We have added a broader impact statement after our conclusion, which emphasizes the role of priors and specific metrics for success in real-world threat models (alongside what the adversary can use, in our case being deterministic given the models returned in training).
>
> **Summary Response to Requested Changes:**
>
> We have provided clarification on Section 4.1 and the term “model utility” in the earlier responses and changed the paper accordingly. We have now also included experiments with more recent MI attacks, where we conclude that they perform worse than LiRA against private models (when measured by positive accuracy).

---

### Review · Reviewer_KveU · 2023-10-31

**Summary Of Contributions:**

This paper studies the membership inference attacks (MIA) against machine learning models, where the training data has been subsampled prior to the training. The main discovery of the paper is that this procedure can significantly lower the MIAs success probability compared to the case where we use all the training data. Moreover, authors present theoretical bounds for the MIA success probability, and show that differential privacy (DP) has smaller effect to these bounds than the subsampling (or equivalently the prior knowledge on target datapoints inclusion in the training data set). Empirical results demonstrate that the positive accuracy of MIA, i.e. probability that MIA correctly classifies target as a member based learned results, decreases significantly when the training data set is subsampled to a smaller set prior to training.

Authors also connect their theoretical results to machine unlearning. The idea is that for a machine unlearning task, curator needs to remove samples from the training data and re-train the model in order to avoid sample being distinguished from the learning outcome. Authors connect this to the probabilistic upper bound for MIAs success. From this bound, authors can derive an upper bound for number of samples the curator can avoid "unlearning" from the model if using the subsampled training data to begin with.

**Audience:**

Yes

**Broader Impact Concerns:**

No broader impact concerns.

**Claims And Evidence:**

Yes

**Requested Changes:**

1. Better diagnostics of the MIA performance. For example, you could add a FPR vs. TPR plots on several different levels of $P_{x^*}(1)$
2. Test how the sampling ratio affects the shadow models. For example you could train a set of shadow models with 10% in 90% out, and see if it has any effect on MIA performance when testing against the $P_{x^*}=0.1$ setting.
3. Explain how $\mathbb{P}(S \mid x^*)$ is computed.

#### After rebuttal
I'm happy with the changes provided by the authors, and have now changed "Claims and Evidence" to "Yes".

**Strengths And Weaknesses:**

## Strengths
Studying how prior knowledge of samples inclusion in the training data set affects membership inference attacks is certainly an interesting one from both theoretical and practical aspects of private machine learning.

The main result of the paper seems to be, that if you subsample your data set prior to the training then the models learned under DP are less susceptible to MI attacks the smaller this subsampling fraction is. Authors demonstrate that this effect is stronger than the MIA reducing effect of DP. This result is highly intuitive, as if a sample is never even used in training, then obviously it should not be susceptible to membership inference.

What is really interesting in this paper is the empirical result that the models performance (i.e. accuracy) is not negatively affected by the smaller training data set. The experiments seem to suggest that there is a "sweet spot" in training set subsampling that for a bounded MIA performance and models accuracy, and that this sweet spot is clearly below 1 (the standard practice of using the entire training data set in training).

## Weaknesses
The term $\mathbb{P}(S \mid x^*)$ in Thm 3.3. is not clear to me. First, I would imagine this term is affected by the privacy guarantee because DP obviously has an effect on the learning methods output distribution. Second, how have you estimated this term in the bounds of Fig 1? I tried looking at the code in the supplementary material but didn't find the script to produce these plots.

I find the positive accuracy, used as a metric throughout the paper, a bit misleading or at least bit incomplete metric. I do understand that small positive accuracy does imply low MI success, however when it is used to diagnose the MI performance with varying $P_{x^*}(1)$ it might have some shortcomings. For example, consider Figure 3. This figure demonstrates that with low sample rate ($P_{x^*}(1)$ is small), the positive accuracy becomes small. More importantly, the change in $P_{x^*}(1)$ has a significantly larger to the positive accuracy than the privacy level $\epsilon$ of DP has. However, this metric is now highly dependent on the strength of the LiRA attack. If the LiRA classifier is unable to learn to distinguish the in/out settings, then the probability of LiRA score exceeding a certain threshold is independent of the in/out status. Now, this means that the number of samples LiRA labels as positive is approximately $N \Pr(s_i > \gamma)$, where $s_i$ is the score for sample $i$ and $\gamma$ is the decision threshold, and if the classifier does not distinguish in/out then $\Pr(s_i > \gamma) = \Pr(s_i > \gamma \mid x_i \in D) = \Pr(s_i > \gamma \mid x_i \notin D)$. Similarly we have that the number of true positives is approximately $N_{pos} \Pr(s_i > \gamma)$, and since $N_{pos} = N P_{x^*}(1)$, we have that true positives / predicted positives is $TP/PP = P_{x^*}(1)$. While it seems that the LiRA classifier used in Fig. 3 is not quite a random classifier (it is somewhat above the diagonal line), it would be really interesting and important to see how powerful this classifier actually is in order to evaluate the difference the $P_{x^*}(1)$ makes. For example in Carlini et al. 2022, the LiRA attack applied on DP models behaved almost like a random classifier even with very moderate noise levels. One interesting question is how would the training data subsampling perform in MIA attacks against non-private models, for which the MIA actually has a high performance.

I would also be quite curious to see how the number of samples used in training affects the MIA performance. In training of shadow models, you have used 50% in/out sets similar to Carlini et al. 2022. However, in your target model, you use training set that might differ in size quite significantly. Would this then affect the MIA performance as the target models could come from quite different distribution compared to the shadow models?

For now, I will set the Claims and Evidence to "No" because of my concerns relating to the positive accuracy as a metric. However, I'm happy to change this if authors can provide include further metrics on the performance of MIA.

References:
Nicholas Carlini, et al., Membership inference attacks from first principles, 2022

## Minor comments/suggestions
- I believe your supplement material is missing some of the plotting scripts, e.g. the one responsible for Fig 1. Moreover, it seems that the plotting script for Fig 3 is bit different to the one actually used in paper (the labels are lower case in attached .py file whereas in the paper the labels are title-case).
- Lemma 3.2: I think you need to assume that D and D' are adjacent (which is done in the proof). Or are you saying that _any_ bijective corresponcence between $D$ and $D'$ would be enough?
- In the proof for Thm 3.3, immediately after " we have the denominator is", are you missing $\Pr(H(D) \in S)$ from the first summand?
- "This will have explained", a typo?
- In Def 3.1., you have used $\mathbb{P}(x^* \in D \mid S)$ do denote the positive prob, but afterwards you use $\mathbb{P}(x^* \mid S)$. I just want to verify that these are the same. If so, it might make sense to add a brief statement after the definition that you will use this notation for the rest of the paper.

---

> ### Author Response · Authors · 2023-12-21
> **Response to Reviewer KveU**
>
> > *The term $P(S| x)$ in Thm 3.3. is not clear to me. First, I would imagine this term is affected by the privacy guarantee because DP obviously has an effect on the learning methods output distribution. Second, how have you estimated this term in the bounds of Fig 1? I tried looking at the code in the supplementary material but didn't find the script to produce these plots.*
>
> To clarify, Figure 1 is actually plotted by taking $\delta = 0$ as described in section 3.4 (i.e., just pure DP) and hence there is no dependence on $P(S|x^*)$. We clarified this in the revised pdf.
>
> More generally, $P(S|x^*)$ is the probability the adversary would accept if we did in fact train on $x^*$, and to use our bound we only need a lower-bound for this value. Hence, what one can do in practice is consider only bounding adversaries that predict positively at least $X$% of the time when we do in fact train on a point; this is in fact necessary as $(\epsilon,\delta)$-DP provides no guarantees for adversaries that predictive positively infrequently as shown in Appendix A.1. In the experiments for Section 4.3 where we set a bound using $\epsilon = 3.0, 2.0, 1.0$ and $P_x(1) = 0.5$ and interpolated across their values, we set our lower-bound on $P(S|x^*)$ to be 1% and now also mention this in Section 4.3 and the relevant figures. In other words, in these experiments we are investigating model performance when providing a positive accuracy bound on the class of all attacks that predict positively at least 1% of the time.
>
>
> > *I find the positive accuracy, used as a metric throughout the paper, a bit misleading or at least bit incomplete metric...One interesting question is how would the training data subsampling perform in MIA attacks against non-private models, for which the MIA actually has a high performance.*
>
> One of the main motivations for positive accuracy is its connection to unlearning; unlearning can be framed as claiming deniability to having trained on a point, equivalent to bounding positive accuracy. Hence, unlearning has a dependence on the TPR vs. FPR curves by the respective DP guarantees they depict. However, the focus of this paper is to show the importance of sampling to bounding positive accuracy (for unlearning) and the interplay with model performance.
>
> Now in terms of our experiments to show that the trend we see theoretically with respect to the prior is somewhat expressed by LiRA, we agree we should also include a TPR vs FPR curve for our attack to understand it's classification ability (not subject to the prior and potentially much worse than what the DP guarantee bounds). We have added results for this in Figure 11 in the revised PDF, and as seen in past work mentioned by the reviewer, LiRA behaves mostly like a random classifier even with weak privacy guarantees. So as mentioned by the reviewer (assuming practically epsilon = 0 privacy and hence indistinguishability), empirically one can expect the positive accuracy to be mostly identical to the sampling probability. However, we want to emphasize that theoretically (against all adversaries in all dataset settings) we cannot guarantee this. Our main experimental result, which is the sweet spot of sampling for performance, is based on the theoretical bound.
>
> We thank the reviewer for suggesting to add experiments on non-private models, and have added results for this in Figure 12 in the revised PDF. We once again see that the positive accuracy improvement of LiRA over the baseline attack varies with sampling rate, as seen in the private training setting.
>
> > *I would also be quite curious to see how the number of samples used in training affects the MIA performance. In training of shadow models, you have used 50% in/out sets similar to Carlini et al. 2022. However, in your target model, you use training set that might differ in size quite significantly. Would this then affect the MIA performance as the target models could come from quite different distribution compared to the shadow models?*
>
> Yes this is interesting, and as suggested have added results for training shadow models with different sampling rates in Figure 13 in the updated PDF. We see no significant difference, though observe that on average $P_{x}(1) = 0.5$ is best. One hypothesis is that this strikes a balance between the amount of  training and non-training data in our setup, so that we can more reliably estimate the two distributions required for LiRA.

---

> > ### Author Response · Authors · 2023-12-21
> > **Response to Reviewer KveU Continued**
> >
> > > *For now, I will set the Claims and Evidence to "No" because of my concerns relating to the positive accuracy as a metric. However, I'm happy to change this if authors can provide include further metrics on the performance of MIA.*
> >
> > We have now also presented the TPR vs. FPR curves for LIRA on the subsampled target models, and described earlier the implications of positive accuracy (hence why it is a metric of interest, though we agree it does not replace other metrics to understand the performance of an adversary). We have also now included further experimentation of positive accuracy in non-private settings, where we once again see a dependence on sampling rates. To summarize, positive accuracy is relevant for certain security concerns (e.g., unlearning), and in this paper we explored its dependence on variables beyond differential privacy.
> >
> >
> > > *I believe your supplement material is missing some of the plotting scripts, e.g. the one responsible for Fig 1. Moreover, it seems that the plotting script for Fig 3 is bit different to the one actually used in paper (the labels are lower case in attached .py file whereas in the paper the labels are title-case).*
> >
> > Refer to our previous response on Figure 1 , which clarifies that it is plotting $1 + e^{-\epsilon}(1- P_x(1)) / P_x(1)$. Nevertheless, we have now also added the plotting scripts for all the numerical experiments and bounds (ours and past work). We have also updated the supplementary material to include the new experiments.
> >
> >
> > > *Lemma 3.2: I think you need to assume that D and D' are adjacent (which is done in the proof). Or are you saying that any bijective corresponcence between $D$ and $D^′$ would be enough?*
> >
> > To clarify $\mathcal{D}$ and $\mathcal{D}’$ are the set of all datasets with and without x^* respectively, and we are stating for this specific pair of sets there is a bijective correspondence based on the associated adjacent datasets D and D’. Furthermore, this bijection comes with a simple map for the probabilities.
> >
> >
> > > *In the proof for Thm 3.3, immediately after " we have the denominator is", are you missing Pr(H(D)∈S) from the first summand?*
> >
> > Yes, thank you it is fixed.
> >
> >
> > **Summary Response to Requested Changes:**
> >
> > As requested, and explained in earlier responses, we have added TPR vs FPR plots for LiRA, and also results for different sampling rates for the shadow models. We have also elaborated how $P(S|x^*)$ was used in this paper in an earlier response (and clarified this in the paper), which is by enforcing a constraint on what adversaries we bound (adversaries that predict positively at least 1\% of the time when training on $x^*$) or by noting that when $\delta = 0$ there is no dependence on $P(S| x^*)$.

---

> > > ### Comment · Reviewer_KveU · 2024-01-25
> > > **After rebuttal**
> > >
> > > Thanks for your very comprehensive response! I now understand the use of $\Pr(S \mid x^*)$ much better, and have no further questions about that. Also thanks for the wide variety of further experiments added. I think they really strenghten your claim.
> > >
> > > I'm happy now to set the claims and evidence to "Yes".
> > >
> > > Minor note: Please label the dashed line in Fig. 11 (I guess it's the y=x?). Also, a loglog plot might be easier to understand.

---

> > > > ### Author Response · Authors · 2024-01-29
> > > > **Response After Rebuttal**
> > > >
> > > > Happy to hear the rebuttal clarified the questions!
> > > >
> > > > On Figure 11, we have updated the caption to clarify what the dashed line is: the $TPR = FPR$ line. For the loglog plots (which we understood to mean log scales for both the x-axis and y-axis), we have plotted them for CIFAR-10 and MNIST when training with $\epsilon = 3$ here: https://anonymous.4open.science/r/TMLR_MI_BOUNDS_Plots-453E. We observed that in our original plots, it is possible to see the slight separation (from the baseline) at low FPR values which is not observable in the loglog plots. If the reviewer still prefers the loglog plots, we can change the paper accordingly.

---

### Review · Reviewer_vVhg · 2023-12-15

**Summary Of Contributions:**

This work derives a bound on the success rate of membership inference attack. The derived bound is claimed to be tighter than the DP bound due to capturing non-DP factors. Additionally, analysis suggests that training on less data can yield more advantageous trade-offs between privacy and utility.

**Audience:**

Yes

**Claims And Evidence:**

No

**Requested Changes:**

Can the authors discuss why their analysis and experiment rely on DP? Especially, it is important to know whether the improvement only exists for DP networks.

**Strengths And Weaknesses:**

The idea of improving the DP bound by considering non-DP factors is interesting. Although the analysis focuses primarily on some specific form of attacks, membership inference can be considered a basic form of privacy attacks, so such analysis is of general interest in the privacy community.

However, I still have some concerns.

i) From Figure 3 I cannot observe what the author has claimed "Sampling is dominant factor dictating how much the LiRA attack outperforms the baseline attack". All the lines appear parallel to the line of the baseline.

iii) From Figure 5 and 6, networks basically collapse for high sampling probability. Does this also suggests the derived bound is still very loose?

---

> ### Author Response · Authors · 2023-12-21
> **Response to Reviewer vVhg**
>
> > *From Figure 3 I cannot observe what the author has claimed "Sampling is dominant factor dictating how much the LiRA attack outperforms the baseline attack". All the lines appear parallel to the line of the baseline.*
>
> To clarify, in this paper we are claiming sampling is “a” dominant factor, and this is best visualized in Figure 4. Figure 4 plots the average improvement of LiRA over the baseline (across the settings) as we vary the training sampling rate, and we see that there is a non-constant dependence with the training sampling rate; as one get closer to 0.5 training sampling rate, the improvement over the baseline increases.
>
> > *From Figure 5 and 6, networks basically collapse for high sampling probability. Does this also suggests the derived bound is still very loose?*
>
> The collapse at high sampling rate is due to the baseline attack of always predicting positive having high positive accuracy (equal to the sampling rate); this implies to retain an a priori fixed bound on positive accuracy, one needs to effectively train with a DP guarantee of $\epsilon \approx 0$, implying the collapse in performance. On the bound being loose, we wish to remark that if for the given task the DP inequalities are tight for every adjacent dataset, then our bounds are tight (see end of Section 3.3). To state in other words, the dependence on the sampling rate is tight.
>
> > *Can the authors discuss why their analysis and experiment rely on DP? Especially, it is important to know whether the improvement only exists for DP networks.*
>
> Our theoretical bound relies on DP as DP is equivalent to ruling out the possibility of observing a model that can only have occurred when training with a point, which leads to the existence of an attack with 100% positive accuracy regardless of sampling rate.
>
> This said, we have now added empirical results on the relation between the positive accuracy of the LiRA attack and the sampling rate of training points for non-DP trained models, presented in Figure 12. We see that for non-DP trained models the improvement over the baseline is a non-constant function of sampling rate, analogous to the DP trained setting. Hence we once again conclude that a dominant factor of the positive accuracy of LiRA (the strongest membership inference attack to the best of our knowledge) is sampling rate even when we do not train with DP.

---

### Decision · Action_Editor_AfqJ · 2024-02-14

**Recommendation:** Accept with minor revision

**Comment:**

## Mathematical Notations:

1. There should be a section about the mathematical notations being used.  This is especially important because some of the notations are not clear and unconventional.  For example, in Definition 3.1 the notation of $\mathbb{P}(x^*\in D | S)$ is used. This typically means a conditional distribution given the event S, where x^* is the random variable.  But to my understanding, in Section 3.1 it means something very different --- it means the probability of an unspecified randomized learning algorithm outputting a model inside S, where S is the set of models determined by a fixed deterministic MI attacker, a fixed dataset $D$ and a fixed target $x^*$.    Specifically, my understanding is that $S = \{ model | MI(x^*, model) = 1 \}$.  For that reason, the standard notation for the MI positive accuracy would be $\mathbb{P}(M([D,x^*]) \in S)$ for a randomized training algorithm M.  Or more directly $\mathbb{P}(MI(x^*, M([D,x^*])) = 1)$ and then specifies that the randomness is only over the randomness of M.


2. It is a bit unclear whether Definition 3.1 is stated for a fixed D or needs to be a bound that applies for any D that includes $x^*$... in the Comment 1 above, I assumed a fixed D that does not include $x^*$ and constructed a particular neighbor dataset $[D,x^*]$.



3. Similarly, the $\mathbb{P}\_{x^*}(0)$ and $\mathbb{P}\_{x^*}(1)$ notations are also a bit awkward too.  The subscript usually denotes the probability distribution and inside the brackets, it is usually the Event.  It is unclear whether it is the probability mass function or the distribution function these notations are about.




## About the first claim:

4. The authors discussed the connection TPR-FPR tradeoff function by citing Kairouz, Oh and Viswanath. It is perhaps better to also cite the origin of this idea Wasserman and Zhou https://arxiv.org/abs/0811.2501, as well as how the mechanism-specific variant known as f-DP from Dong, Roth and Su https://arxiv.org/abs/1905.02383 could be used to improve the bounds.  Specifically, the authors used generic DP definition for their analysis, but DP-SGD satisfies RDP and f-DP, which implies stronger protection against membership inference.


5. The authors should verify that their Theorem 3.3 is not already implied by Theorem 4.2 and Corollary 4.3 from Dong, Roth and Su https://arxiv.org/abs/1905.02383.   Note that Type I error = 1 - Positive accuracy  and Type II error =  1- Negative accuracy.



## Now about the second claim:

6. So there are two sampling procedures in this paper. The first, is the poisson sampling of the minibatches within DP-SGD. The second is another poisson sampling (deccribed in the lasts paragraph of Section 3.2 before Lemma 3.2 that happens before the data is passed into DP-SGD.  I am curious that in the experiments (e.g., the $\epsilon$ in the legends of Figure 1 and Figure 3), has the reported DP parameter \epsilon taken advantage of the amplification by sampling that happened at the beginning?

7. In Figure 1 for example, it makes sense to alsso include the Pos Acc bound directly implied by the DP guarantee of the mechanism.

**Audience:**

The reviewers and I all find this paper of interest to the TMLR readers.  In particular,  researchers and practitioners in differential privacy, differentially private machine learning,  privacy auditing, and privacy attacks may all find this paper relevant.

**Claims And Evidence:**

The paper primarily makes two claims.

1. A bound on the "positive accuracy" and "negative accuracy" of membership inference (MI) attacks for a learning algorithm satisfying approximate DP  (e.g., DP-SGD). The bound improves existing bounds that do not incorporate a "prior".

2. The subsampling probability has a strong influence in the success of membership inference attacks (and there is a sense that it provides more influence than the DP parameter).

The reviewers, especially KveU, have validated that correctness of the proofs on the formal statements.  The empirical claim were initially not clear but the authors substantially improved the experimental session which now tells a more convincing story to support Claim 2 above.

I have some additional comments and questions that the authors need to confidently address in the "comments" section later before I can confidently approve the minor revision.

---

> ### Author Response · Authors · 2024-02-17
> **Response to Comments**
>
> We thank the reviewers for all the valuable feedback. We now respond to the additional comments raised by the Action Editor.
>
>
> >There should be a section about the mathematical notations being used. This is especially important because some of the notations are not clear and unconventional. For example, in Definition 3.1 the notation of $\mathbb{P}(x^* \in D| S)$ is used...
>
> Yes, we have added a clarification of this notation into the paper (alongside a more substantial math notation paragraph in Section 3.1 before the definition of MI positive accuracy). To be clear, it is not the probability that the MI attack predicts positively over the randomness of M (and D), but the probability that given the MI attack predicts positively, x was in D. So the randomness is over the sampling of D (the random variable) and can be written formally as $\mathbb{P}(D \in D_{x^*} | MI(x^*, M(D)) = 1)$ where $D_{x^*}$ is the event space of datasets containing $x^*$.
>
> >It is a bit unclear whether Definition 3.1 is stated for a fixed D or needs to be a bound that applies for any D that includes $x^*$...
>
> As hopefully clarified above, the random variable of concern is $D$ which follows i.i.d sampling from some larger set of points. So it is neither any D containing $x^*$ nor a specific $D$, but in fact a statement about the probability of the random variable $D$ belonging to the event space of datasets containing $x^*$ ($\mathcal{D}_x^*$) conditioned on the MI attack on $x^*$ given M(D) being positive.
>
> >Similarly, the $P_{x^*}(1)$ and $P_{x^*}(0)$ notations are also a bit awkward too...
>
> We have also clarified this in the math notation paragraph in Section 3.1, where we use it to denote the bernoulli random variable of $x^* \in D$ (where again the dataset $D$ is the random variable of concern) with $\mathbb{P}_{x^*}(1)$ being the probability of it being 1 (representing $x^*$ is sampled in $D$). Note $D$ can then be defined as the product space of the bernoulli random variables for each $x$.
>
> >The authors discussed the connection TPR-FPR tradeoff function by citing Kairouz, Oh and Viswanath. It is perhaps better to also cite...
>
> We agree and have added discussion on future work using RDP and f-DP for tighter bounds in Section 2.1 on DP. We now also mention Wasserman and Zhou in the caption of Figure 11.
>
> >The authors should verify that their Theorem 3.3 is not already implied by Theorem 4.2 and Corollary 4.3 from Dong, Roth and Su https://arxiv.org/abs/1905.02383. Note that Type I error = 1 - Positive accuracy and Type II error = 1- Negative accuracy.
>
> We have looked at the bound and note that first the sampling procedure is different (they are sampling uniformly among subsets of size m for some fixed m, while we use i.i.d sampling of individual points). Secondly, as hopefully is clearer from the discussion on Definition 3.1, our hypothesis test is different than f-DP as we are letting $D$ be a random variable and asking when it lies in $\mathcal{D}_x^*$, and not which of a specific pair of neighbouring datasets was used. Hence converting their bound to our bound is not immediate, as it requires relating the hypothesis test on neighboring datasets to now having the dataset be a random variable and asking when it is in $\mathcal{D}_x^*$, which is the core contribution of Theorem 3.3 (which does this conversion for $(\epsilon,\delta)$-DP). A direction for future work is to do the same conversion with f-dp and see if it leads to quantitatively different bounds for DP-SGD.
>
>
> >So there are two sampling procedures in this paper... I am curious that in the experiments (e.g., the $\epsilon$ in the legends of Figure 1 and Figure 3), has the reported DP parameter $\epsilon$ taken advantage of the amplification by sampling that happened at the beginning?
>
>
> No it has not, and this is as we have separated the role of $D$ being a random variable due to i.i.d sampling to how the training batch sampling improves indistinguishability between specific neighboring datasets. More formally, DP sampling analyzes the difference between M(Sample(D)) and M(Sample(D’)) where $D$ and $D’$ are fixed neighboring subsets of \{x_1,\cdots,x_n\}, and our dataset sampling defines $D$ as a random variable sampled from $\{x_1,\cdots,x_N\}$. To use the first sampling for DP would mean to analyze the DP guarantee between two different adjacent pools of possible data used to train $\{x_1,\cdots,x_N\}$ and $\{x_1,\cdots ,x_{N},x_{N+1}\}$ which is not the objective of our bounds (we care about the specific training datasets that were sampled from a pool of data).

---

> > ### Author Response · Authors · 2024-02-17
> > **Response to Comments Continued**
> >
> > >In Figure 1 for example, it makes sense to alsso include the Pos Acc bound directly implied by the DP guarantee of the mechanism.
> >
> > As hopefully now clearer from the discussion of Theorem 4.2 from Dong, Ruth, and Su and the difference between the hypothesis test of DP and our hypothesis test, there is no positive accuracy guarantee implied by the DP guarantee without defining the random variable $D$. More generally, positive accuracy changes the hypothesis test of DP to the hypothesis test that the training dataset $D$ (the random variable of concern) belongs to set of datasets containing $x^*$ given $MI(x^*, M(D)) = 1$. Figure 1 presents our bound on the $1- \text{Type-1}$ Error of this hypothesis test given $M$ is DP and $D$ is defined by i.i.d sampling datapoints from a pool $\{x_1,\cdots,x_N\}$ with individual probabilities given on the x-axis.